# Characterisation of a nucleo-adhesome

Adam Byron [1,4✉], Billie G. C. Griffith[1,9], Ana Herrero [1,5,9], Alexander E. P. Loftus[1], Emma S. Koeleman[1,2,6], Linda Kogerman[1], John C. Dawson [1], Niamh McGivern[1,7], Jayne Culley[1], Graeme R. Grimes[3], Bryan Serrels[1,8], Alex von Kriegsheim[1], Valerie G. Brunton[1] & Margaret C. Frame [1]

In addition to central functions in cell adhesion signalling, integrin-associated proteins have wider roles at sites distal to adhesion receptors. In experimentally defined adhesomes, we noticed that there is clear enrichment of proteins that localise to the nucleus, and conversely, we now report that nuclear proteomes contain a class of adhesome components that localise to the nucleus. We here define a nucleo-adhesome, providing experimental evidence for a remarkable scale of nuclear localisation of adhesion proteins, establishing a framework for interrogating nuclear adhesion protein functions. Adding to nuclear FAK's known roles in regulating transcription, we now show that nuclear FAK regulates expression of many adhesion-related proteins that localise to the nucleus and that nuclear FAK binds to the adhesome component and nuclear protein Hic-5. FAK and Hic-5 work together in the nucleus, co-regulating a subset of genes transcriptionally. We demonstrate the principle that there are subcomplexes of nuclear adhesion proteins that cooperate to control transcription.

[1] Cancer Research UK Edinburgh Centre, Institute of Genetics and Cancer, University of Edinburgh, Edinburgh EH4 2XR, UK. [2] Leiden University Medical Center, 2333 ZC Leiden, The Netherlands. [3] MRC Human Genetics Unit, Institute of Genetics and Cancer, University of Edinburgh, Edinburgh EH4 2XU, UK. [4] Present address: Division of Molecular and Cellular Function, School of Biological Sciences, Faculty of Biology, Medicine and Health, University of Manchester, Manchester M13 9PT, UK. [5] Present address: Instituto de Biomedicina y Biotecnología de Cantabria (IBBTEC), Consejo Superior de Investigaciones Científicas (CSIC)-Universidad de Cantabria, 39011 Santander, Spain. [6] Present address: Division of Chromatin Networks, German Cancer Research Center (DKFZ) and Bioquant, 69120 Heidelberg, Germany. [7] Present address: Almac Diagnostic Services, 19 Seagoe Industrial Estate, Craigavon BT63 5QD, UK. [8] Present address: NanoString Technologies, Inc., Seattle, WA 98109, USA. [9] These authors contributed equally: Billie G. C. Griffith, Ana Herrero. ✉email: adam.byron@ed.ac.uk

Numerous intracellular signalling, adaptor and cytoskeletal proteins associate with transmembrane integrin receptors in adhesion complexes, which transduce biochemical and biomechanical signals to integrate the extracellular matrix (ECM) with the actomyosin cytoskeleton[1–3]. Adhesion complex-associated proteins, collectively termed the adhesome, form highly dynamic and complex functional interaction networks, and their spatiotemporal control plays an important role in intracellular signalling and biological processes such as cell proliferation and migration[4–7].

Development of methods for the biochemical isolation and mass spectrometric quantification of adhesion complex-associated proteins has enabled the proteomic and bioinformatic characterisation of adhesome networks[8–14]. Integration in silico of multiple fibronectin-based adhesion-site subproteomes from different cell types defined a meta-adhesome, from which a core set of 60 frequently identified adhesion proteins (a consensus adhesome) was identified[15]. In addition, a literature-curated adhesome database documents over 200 adhesion complex-associated proteins reported in studies that used a variety of cell types, methodologies and experimental conditions[16,17].

Many of the proteins in the consensus adhesome have well-understood structural or signalling functions at regions of cell-ECM and, in some cases, cell-cell adhesion. Yet studies in various cellular systems have, in some instances, described the localisation of certain adhesion proteins at sites distal to the cell-ECM interface. For example, zyxin, paxillin, α-actinin, TRIP6 and focal adhesion kinase (FAK) have been reported to localise to the nucleus under conditions associated with cellular stress (reviewed in refs. [4,18–21]). For FAK, we, and others, have demonstrated nuclear localisation and molecular and biological functions[22–28]. In some cases, these can be related to oncogenic stress; for example, FAK is not detectable in nuclear fractions of normal keratinocytes, but it accumulates in the nucleus of their malignant squamous cell carcinoma (SCC) counterparts[25]. Nuclear FAK is important, since we have shown that it drives tumour immune evasion and the growth of tumours in vivo by building molecular transcription complexes in the nucleus that regulate the expression of cytokines, such as IL-33 and Ccl5 (refs. [25,27,28]). However, in contrast to all that we know for FAK, very little is known about the true extent of nuclear localisation and functions of adhesion proteins.

Here, we provide a comprehensive study using subcellular proteomics and network analysis to characterise a nucleo-adhesome in mouse SCC cells that were used previously to establish a critical role for nuclear FAK[25]. Surprisingly, more than half of proteins in the consensus adhesome localise to the nucleus in SCC cells, implying that the translocation of classical adhesion proteins to the nucleus is common. Our data (i) enable a quantitative, systems-level analysis that extends knowledge of which adhesion proteins can translocate to the nucleus and (ii) establish a framework for developing a fuller understanding of the extent and functions of nucleo-adhesome proteins in the nucleus. We advance previous findings on the well-understood, vital role for nuclear FAK by demonstrating that nuclear FAK transcriptionally regulates the expression of a subset of adhesion proteins that can localise to the nucleus. We show that nuclear adhesion proteins can co-associate and co-regulate a subset of genes via transcription, as exemplified by a FAK–Hic-5 nuclear complex.

## Results

**Enrichment of nucleus-localised proteins in the meta-adhesome.** Mass spectrometric analyses of adhesion complex-associated proteins have indicated that cell-ECM adhesions are sites of greater molecular complexity and diversity than previously appreciated[29–33]. Since there is no reason why cellular proteins only have a single subcellular location, we hypothesised that the molecular composition of the adhesome could reveal functional links to putative alternative roles or subcellular distributions of adhesion-localised proteins. To examine the subcellular provenance of adhesion proteins, we used bioinformatic approaches to interrogate the meta-adhesome, a proteomics-based database of adhesion complex-associated proteins derived from multiple cell types[15]. As expected, the meta-adhesome was enriched for proteins involved in integrin signalling, cell junctions, protein trafficking and the cytoskeleton (Fig. 1a, b, Supplementary Data 1). However, we also noted over-representation of adhesion complex-associated meta-adhesome proteins that have been reported at noncanonical subcellular locations, suggesting roles in processes other than cell adhesion, and a substantial proportion of these were associated with the nucleus (including the spliceosome, small nucleolar ribonucleoprotein complex, ribosome and chromosome) (Fig. 1a, b, Supplementary Data 1).

We used functional network analysis to assess the overlap of subcellular regions or biological processes enriched in the meta-adhesome (Fig. 1c, d, Supplementary Fig. 1a, b). Topological analysis of graph-based clustering revealed connectivity (constituent protein overlap) between adhesion-related compartments and the nucleus (e.g. cytoskeleton and nucleoplasm) (Fig. 1c, Supplementary Fig. 1b), indicating meta-adhesome proteins reported at both adhesion-relevant and nuclear locales. Furthermore, graph-based clustering of enriched functional categories defined functional links from cell adhesion (orange nodes, Fig. 1d) to intracellular transport (including nucleocytoplasmic transport) (green nodes, Fig. 1d) and nuclear processes (purple nodes, Fig. 1d) (Supplementary Fig. 1a). Indeed, 42.3% (30 proteins) of known LIM (Lin11–Isl1–Mec3) domain-containing proteins, which can shuttle between the cytoplasm and the nucleus[34], are found in the meta-adhesome (false-discovery rate (FDR)-adjusted $P = 6.04 \times 10^{-8}$, hypergeometric test with Benjamini–Hochberg correction; Supplementary Data 1), of which half (15 proteins) are members of the core consensus adhesome[15] and form a network of physical and functional associations (Supplementary Fig. 1c). Furthermore, in addition to the anticipated enrichment of proteins that can interact with adhesion-related molecules such as fibronectin and α4 integrin ($P = 0$ and $1.55 \times 10^{-179}$, respectively; Supplementary Data 1), there was an over-representation of meta-adhesome proteins that have been reported to interact with non-adhesome nuclear proteins such as the chromatin-modifying polycomb protein EED, the histone deacetylase sirtuin-7, the perinuclear cytoskeletal protein obscurin-like 1 and the R2TP cochaperone complex subunit PIH1D1 ($P = 0$, $6.38 \times 10^{-197}$, $2.64 \times 10^{-190}$ and $1.77 \times 10^{-183}$, respectively; Supplementary Data 1). These analyses imply that a subset of adhesion proteins can localise to the nucleus and potentially associate with nuclear proteins, and we postulated that this may represent a previously unappreciated nucleo-adhesome.

**Subcellular proteomics characterises a nucleo-adhesome.** To interrogate the nuclear localisation of adhesion proteins, we enriched nuclei from SCC cells that we used previously to decipher nuclear functions for FAK[25–28] for biochemical analysis. As organelles contiguous with or adjacent to the nuclear membrane, such as the endoplasmic reticulum (ER), often contaminate nuclear preparations[35], we developed a method that optimised removal of perinuclear cellular components. We used a subcellular fractionation approach based on adaptation of a previously reported isotonic buffer-mediated cellular dissection

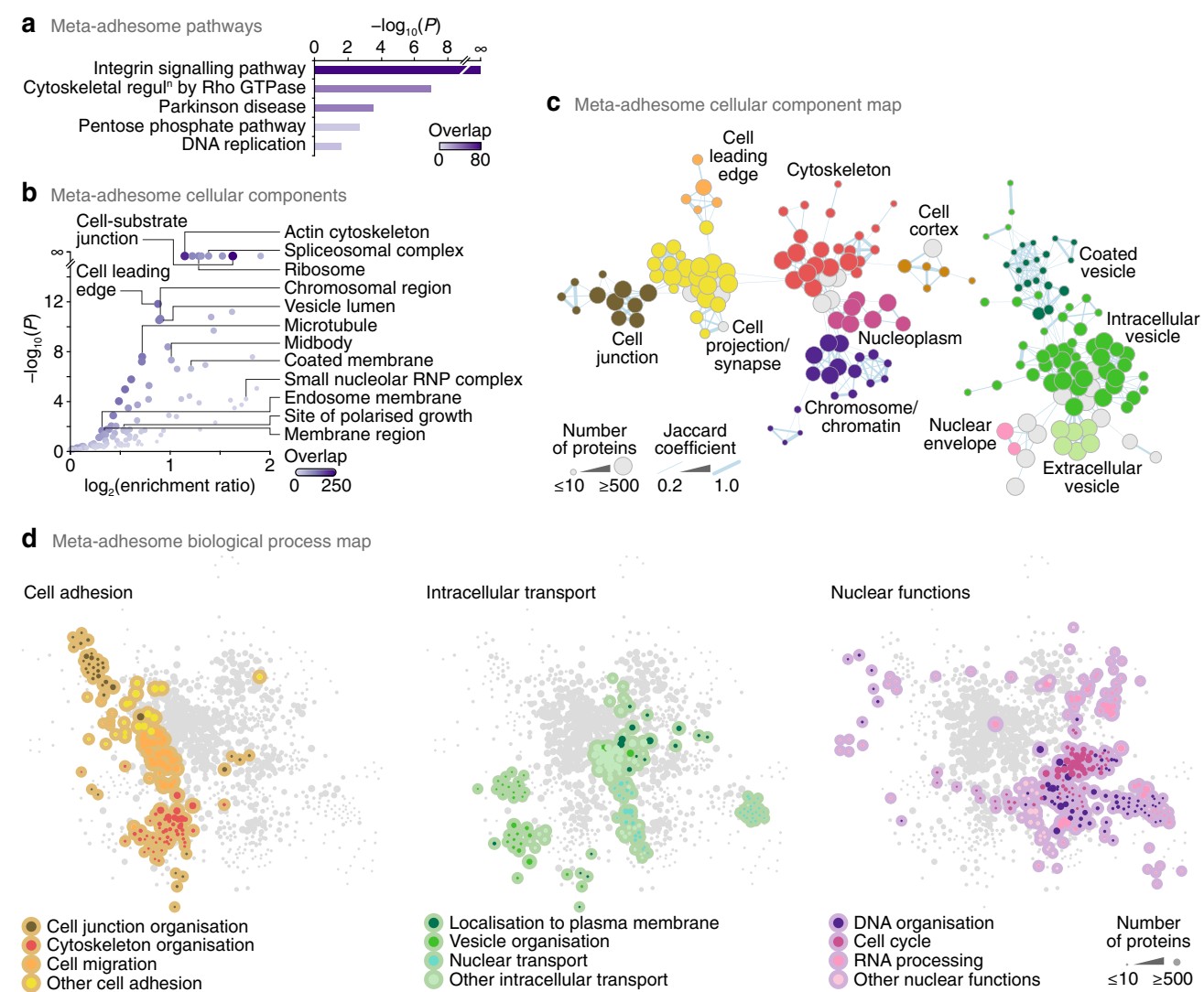

**Fig. 1 Nucleus-associated adhesome proteins. a**, **b** Over-representation analyses of PANTHER pathways (**a**) and Gene Ontology cellular components (**b**) in the meta-adhesome. Representative labelled terms were determined using affinity propagation, except for the terms cell-substrate junction and cell leading edge. Purple shading intensity indicates size of meta-adhesome protein overlap with respective gene sets. For **a**, terms with enrichment ratio (observed genes/expected genes in gene set) > 2 are displayed ($P < 0.025$, one-sided hypergeometric test with Benjamini–Hochberg correction). RNP, ribonucleoprotein. **c** Graph-based clustering of over-represented cellular components in the meta-adhesome. A force-directed graph was generated from gene-set membership of enriched Gene Ontology terms ($P < 0.01$, one-sided hypergeometric test with Benjamini–Hochberg correction). Representative cellular components summarising selected clusters of enriched terms are labelled. Node (circle) size represents the number of meta-adhesome proteins annotated for each enriched term; node fill colour represents cellular component annotation. Edge (line) weight is proportional to the overlap of gene-set membership of connected nodes (Jaccard coefficient ≥ 0.2). The two largest connected components are shown (see also Supplementary Fig. 1b). **d** Graph-based clustering of over-represented biological processes in the meta-adhesome. Enriched Gene Ontology terms associated with cell adhesion (left panel), intracellular transport (middle panel) and nuclear functions (right panel) are coloured as indicated. The largest connected component is shown, and edges were omitted for clarity (see Supplementary Fig. 1a).

technique[36] (Fig. 2a). We found that the resulting nuclear fractions were depleted of protein markers of the plasma membrane, cytoplasm, organellar membrane and perinuclear region but retained nuclear proteins (Fig. 2b, Supplementary Fig. 2a). Undetectable α-tubulin and actin filament-associated protein 1 in the isolated nuclear fractions implied that potential nonspecific accumulation of some cytoskeletal components was minimised using this method (Fig. 2b, Supplementary Fig. 2a). Together, these data indicate, and provide confidence, that non-nuclear organellar and nonspecific membrane-associated proteins were not enriched in the nuclear preparations.

We used quantitative proteomics to analyse in depth the extent of nuclear localisation of adhesion proteins. Analysis of isolated subcellular fractions using liquid chromatography-coupled tandem mass spectrometry (LC-MS/MS) (Fig. 2c) identified 509,344 peptide-spectrum matches, from which 5,016 protein groups were quantified in at least four out of five biological replicate experiments (Supplementary Data 2). Biological replicate fractions were well correlated (Spearman rank correlation coefficient (ρ) ≥ 0.91, $P = 0$, Spearman's test; Supplementary Fig. 2b). A proportion of proteins were detected in multiple subcellular locations (Fig. 2d, Supplementary Fig. 2c), which may, in part, reflect the propensity for many proteins in the proteome to localise to, and function at, multiple subcellular locales[37,38]. However, the quantified subproteomes were clearly distinguishable in principal component space (Fig. 2e) and formed separate

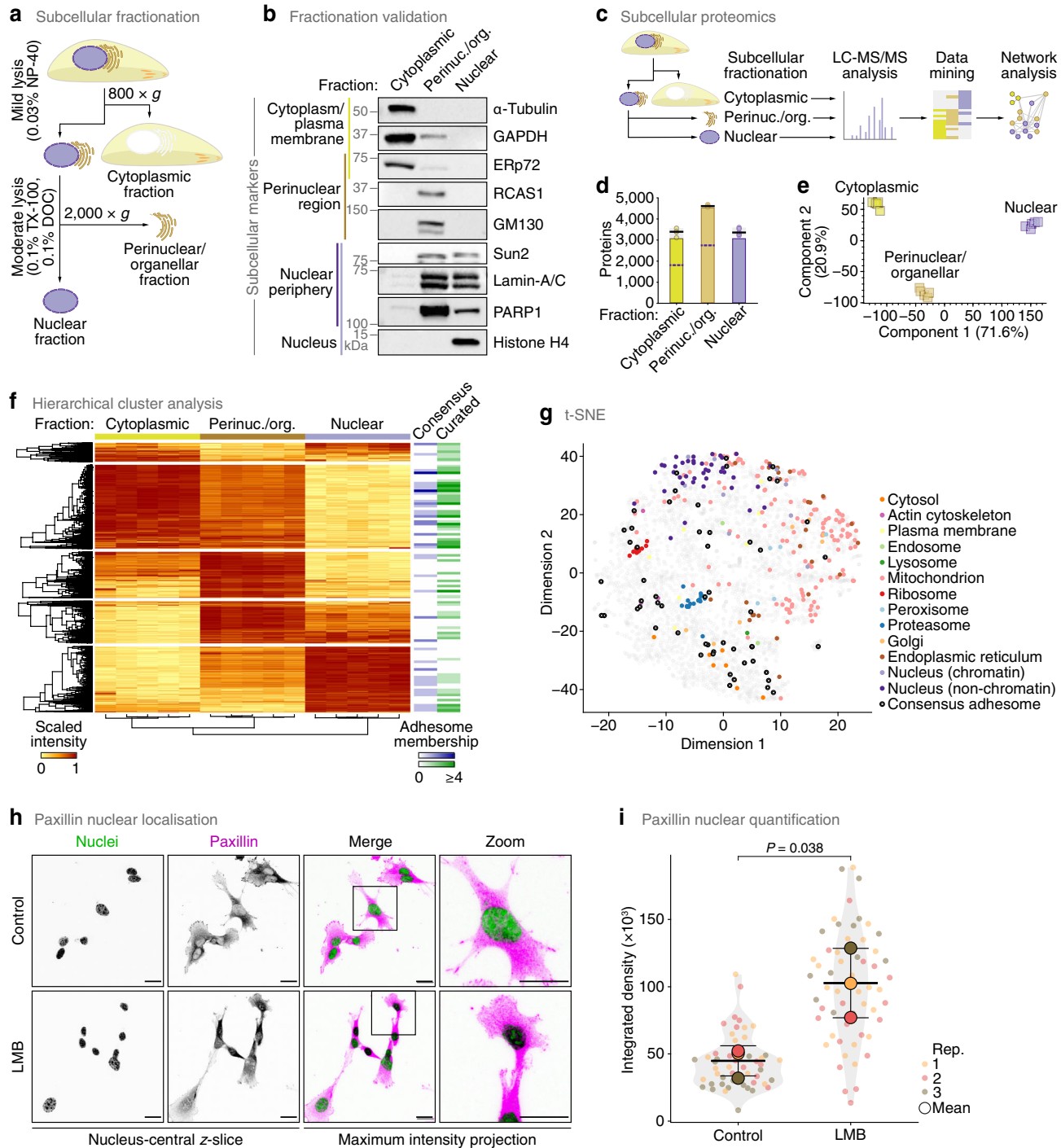

**a** Subcellular fractionation

**b** Fractionation validation

**c** Subcellular proteomics

**d**

**e**

**f** Hierarchical cluster analysis

**g** t-SNE

**h** Paxillin nuclear localisation

**i** Paxillin nuclear quantification

clusters based on protein abundance (Fig. 2f) and intersample correlation (Supplementary Fig. 2b), confirming that the different subcellular fractions were quantitatively distinct. Indeed, the nuclear subproteome was defined by a distinct cluster of nucleus-enriched proteins that was depleted of cytoplasmic proteins (Fig. 2f), and subcellular marker proteins generally partitioned according to subcellular location (Fig. 2g, Supplementary Fig. 2d). Some mitochondrial proteins overlapped with the set of nuclear marker proteins in t-SNE space (Fig. 2g), possibly reflecting the reported localisation of a subset of mitochondrial proteins to the nucleus[39] or minor contamination with some mitochondrial proteins. We used support vector machine (SVM)-based supervised learning to classify proteins according to the partitioning of subcellular marker proteins (Supplementary Fig. 2e, f, Supplementary Data 2). Notably, many adhesome components were present in the cluster of nucleus-enriched proteins (Fig. 2f, Supplementary Data 2), with several of these partitioning with predicted nucleus-resident proteins (Supplementary Fig. 2f, Supplementary Data 2), implying that a subset of adhesion proteins was quantitatively enriched in SCC cell nuclei.

To verify nuclear adhesion protein localisation using an exemplar, we quantified the enrichment of the adhesion protein paxillin in the nucleus of these cells using immunofluorescence and confocal microscopy (Fig. 2h). Paxillin was detectable in the nucleus under control conditions, but inhibition of exportin-1

**Fig. 2 Characterisation of a nucleo-adhesome. a** Methodological workflow for subcellular fractionation and enrichment of nuclei. DOC, sodium deoxycholate; TX-100, Triton X-100. **b** Effective subcellular fractionation of SCC cells. Markers for given subcellular locations are indicated. Immunoblots are representative of five independent experiments. **c** Workflow for mass spectrometric characterisation of cytoplasmic, perinuclear and nuclear subproteomes of SCC cells. **d** Numbers of proteins identified in each subcellular fraction. Black bar, median; light grey box, range; circle, replicate data point ($n = 5$ independent biological replicates). Numbers of proteins identified in at least four out of five biological replicate subcellular fractions are represented as bars. Purple dashed lines indicate numbers of proteins also identified in nuclear fractions (regardless of relative abundance). **e** Principal component analysis of proteins quantified in at least four out of five biological replicate experiments. **f** Hierarchical cluster analysis of the cytoplasmic, perinuclear and nuclear subproteomes. Relative protein abundance was min-max scaled protein-wise (scaled intensity). Memberships of the consensus adhesome and literature-curated adhesome are indicated (bins, 50 proteins). **g** t-SNE map of the subcellular proteomes (Supplementary Data 2) annotated with curated subcellular markers and consensus adhesome proteins. **h** Confocal imaging of SCC cells in the presence or absence of 10 nM leptomycin B (LMB). Nuclei were detected using NucBlue. z-slices passing through the centres of the nuclei (greyscale channels) are shown alongside maximum intensity projections (merged channels). Inverted lookup tables were applied; in merged images, colocalisation of paxillin (magenta) and NucBlue (green) is represented by black regions. Images are representative of three independent experiments. Scale bars, 20 μm. **i** Quantification of nuclear paxillin signal in nucleus-central z-slices (see **h**). Black bars, condition mean (thick bar) ± s.d. (thin bars); grey silhouette, probability density. Data from different biological replicates (rep.) are indicated by coloured circles; replicate means are indicated by large circles. Statistical analysis, two-sided Welch's t-test of replicate means ($n = 52$ and 57 cells for control and LMB treatment, respectively, from $n = 3$ independent biological replicates). Source data are provided as a Source Data file.

using leptomycin B resulted in strong nuclear, but not nucleolar, accumulation of paxillin (Fig. 2i).

**Network analysis identifies canonical and noncanonical nucleo-adhesome components.** We next interrogated the composition of the nuclear subproteome of SCC cells. Strikingly, we found that 71.3% (1,719 proteins) of the experimentally derived meta-adhesome[15] was detected in nuclear fractions, with 1,212 meta-adhesome proteins (50.2%) stringently identified in all five biological replicate experiments (Supplementary Data 3). For further analysis and thresholding of the dataset, we classified proteins identified in all five experiments, and quantified with a > 5% fraction of the cellular pool, as high-stringency identifications. Meta-adhesome proteins identified in nuclear fractions with high stringency were enriched for proteins with nuclear functions and known nuclear localisation as compared to the total set of meta-adhesome components (Supplementary Data 4). While many meta-adhesome proteins were detected at relatively low abundance in the nucleus, a substantial proportion of high-stringency nuclear meta-adhesome proteins had high relative abundance in nuclear fractions compared to the other subcellular fractions (Supplementary Fig. 3a). Moreover, 56.7% (34 proteins) of the core consensus adhesome[15] was robustly identified in our high-stringency nuclear subproteome, and we detected 64 proteins (27.6%) of the literature-curated adhesome[17] (Fig. 3a, Supplementary Fig. 3b). These data establish a subset of the adhesome that can localise to the nucleus, which we termed a nucleo-adhesome, here defined using SCC as a cellular model.

We defined a 34-protein core nucleo-adhesome consisting of components of the consensus adhesome identified in nuclei with high stringency (Supplementary Data 3). We then used interaction network analysis to model the nuclear adhesion proteins based on the putative signalling axes (modules) derived from the curated consensus adhesome network[15] as well as those that were not assigned a consensus adhesome module (Fig. 3b, Supplementary Fig. 3c). This resulted in a core nucleo-adhesome network that included many of the adhesion proteins that had been previously reported to shuttle to the nucleus[4,18–21], providing confidence that our approach captured known nuclear adhesion proteins.

The three FERM (four-point-one, ezrin, radixin, moesin) domain-containing proteins of the consensus adhesome, kindlin-2, talin-1 and FAK, were detected in nuclear fractions, with FAK identified with high stringency (i.e. in all five biological replicate experiments, quantified with >5% in the nuclear pool). Indeed, 76.5% of the FERM domain-containing proteins in the meta-

adhesome (13 out of 17 proteins) were detected in nuclear fractions (9 proteins with high stringency) (Supplementary Data 3). While the extent of this was surprising, FERM domain-containing proteins are frequently implicated in linking the plasma membrane to the cytoskeleton and many have predicted nuclear export signals, with several known to localise to the nucleus[40].

Half of the consensus adhesome components that bind the actin cytoskeleton (11 proteins) were identified in nuclei with high stringency; predominantly, these were from the actin-binding-rich α-actinin adhesome module[15] (Fig. 3b, Supplementary Fig. 3c). Several LIM domain-containing consensus adhesome proteins, most of which can interact directly or indirectly with the actin cytoskeleton, were identified with high stringency in the nucleo-adhesome (Fig. 3b). All of these core nucleo-adhesome proteins have been previously described to translocate to the nucleus, including FHL2 and 3 (refs. [41,42]), LIM and SH3 domain protein 1 (LASP1) (ref. [43]), PDZ and LIM domain protein (PDLIM)1 and 7 (ref. [44]) and TRIP6 (ref. [45]). To explore additional LIM domain-containing adhesion proteins in the nucleus, we expanded the core nucleo-adhesome network to include connected consensus adhesome proteins detected in nuclear fractions without high stringency (Fig. 3c). The expanded network included the known nuclear-localised LIM domain-containing consensus adhesome proteins LPP, paxillin (Fig. 2h), PINCH-1 and zyxin[46–49]. Interestingly, we also detected the LIM domain-containing protein testin – a focal adhesion protein and tumour suppressor that has not, to our knowledge, been previously reported to function in the nucleus (Fig. 3c, d, Supplementary Data 3). Testin associated with other actin-binding nucleo-adhesome components in the network based on previously reported protein interactions (Fig. 3d). We confirmed the presence of testin in nuclear subcellular fractions using immunoblotting (Supplementary Fig. 3d). Using immunofluorescence and confocal microscopy, we found that testin is detectable in the nucleus under control conditions and accumulates in the nucleus in response to hydrogen peroxide-induced oxidative stress (Fig. 3e, f), which is known to regulate the nuclear localisation of several transcription regulators and adhesion-related proteins[19,50], indicating the capacity of our dataset to serve as a proteome-scale resource for the discovery of previously unappreciated nuclear adhesion proteins. These systems-level data support the notion that LIM domain-containing adhesion proteins shuttle between the cell surface and the nucleus[19] and extend the complement of adhesion-associated LIM proteins reported to localise to the nucleus.

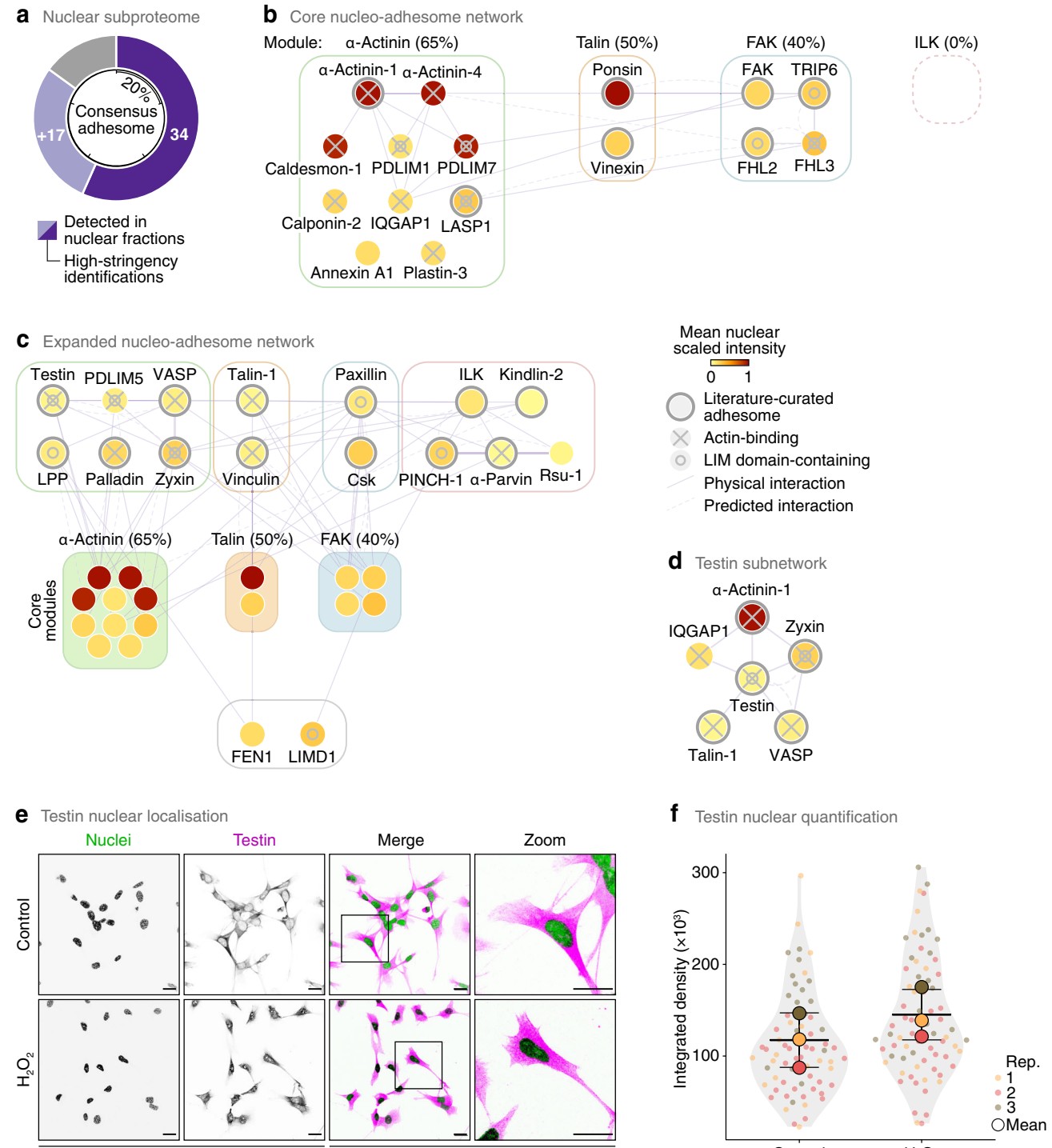

**a** Nuclear subproteome

**b** Core nucleo-adhesome network

**c** Expanded nucleo-adhesome network

**d** Testin subnetwork

**e** Testin nuclear localisation

**f** Testin nuclear quantification

A substantial proportion of consensus adhesome components that were unconnected to or unassigned a network module (83.3%) were identified in nuclei with high stringency (Supplementary Fig. 3c). While many of the nuclear adhesion proteins in the core adhesome modules were quantitatively enriched in non-nuclear fractions (yellow nodes, Fig. 3b), proteins not assigned an adhesome module were almost all strongly enriched in the nuclear fraction (red nodes, Supplementary Fig. 3c). One possibility is that these unassigned consensus adhesome components, which are frequently detected in adhesion complexes but poorly understood in the context of canonical adhesion functions, may have their major roles in the nucleus and

may exist in relatively low abundance, or have moonlighting functions, in non-nuclear locales such as at cell adhesions. In support of this, experimental evidence from a variety of cell types has demonstrated the nuclear localisation of all proteins not assigned an adhesome module, with cell membrane or cytoplasmic localisation also reported for the majority of these proteins (Supplementary Fig. 3e). As examples from our network, the Rap1 GTPase-activating protein Sipa-1 (signal-induced proliferation-associated protein 1), which negatively regulates cell-substrate adhesion, is predominantly perinuclear and nuclear as well as cytoskeletal[51-53]. Heterogeneous nuclear ribonucleoprotein (hnRNP) Q, which regulates adhesion complex

**Fig. 3 Network analysis identifies testin as a nucleo-adhesome-associated protein. a** Proportion of the consensus adhesome quantified in SCC cell nuclear fractions. High-stringency proteins (dark purple segments) were identified in all five biological replicate experiments and quantified with a > 5% fraction of the cellular pool; light purple segments indicate additional proteins detected in nuclear fractions. Tick marks indicate 20% increments. **b** Curated interaction network model of high-stringency nuclear proteins present in the four putative signalling axes (modules) of the consensus adhesome. Coverage of each module is indicated in parentheses. **c** Curated core nucleo-adhesome network (see **b**) expanded to include additional consensus adhesome proteins detected In nuclear fractions. **d** Direct interaction neighbourhood of testin in the expanded nucleo-adhesome network (see **c**). For **b**–**d**, node (circle) fill colour represents the mean scaled intensity of nuclear fraction replicates; thick grey node borders indicate representation in the literature-curated adhesome. Edges (lines) represent reported interactions. **e** Confocal imaging of SCC cells in the presence or absence of hydrogen peroxide ($H_2O_2$). Nuclei were detected using NucBlue. z-slices passing through the centres of the nuclei (greyscale channels) are shown alongside maximum intensity projections (merged channels). Inverted lookup tables were applied; in merged images, colocalisation of testin (magenta) and NucBlue (green) is represented by black regions. Images are representative of three independent experiments. Scale bars, 20 μm. **f** Quantification of nuclear testin signal in nucleus-central z-slices (see **e**). Black bars, condition mean (thick bar) ± s.d. (thin bars); grey silhouette, probability density. Data from different independent biological replicates (rep.) are indicated by coloured circles; replicate means are indicated by large circles. $P > 0.05$; statistical analysis, two-sided Welch's t-test of replicate means ($n = 67$ and $72$ cells for control and $H_2O_2$ treatment, respectively, from $n = 3$ independent biological replicates). Source data are provided as a Source Data file.

formation via RhoA signalling, can localise to the nucleus and the cytoplasm[54]. With little known about their roles in cell adhesion, the DEAD-box helicases DDX18 and DDX27 and the ribosome biogenesis protein Brx1 bind ribosomal RNA in the nucleolus[55–57], and ALYREF (also known as THO complex subunit 4) is an essential nuclear messenger RNA (mRNA) export factor and transcriptional coactivator[58]. Taken together, our data define a nucleo-adhesome, and this is populated not only with actin-binding and LIM domain-containing proteins, but also with noncanonical core adhesion proteins, implying unappreciated nuclear roles for many adhesion proteins.

**Analysis of the FAK-dependent nuclear proteome**. Given the now well-documented and important nuclear functions of the adhesion protein FAK[23–28], which is a component of the core nucleo-adhesome network we describe here (Fig. 3b), we examined whether there was a role for FAK in specifying the nucleo-adhesome and, more broadly, the nuclear proteome. We confirmed the nuclear localisation, and chromatin association, of wild-type FAK (FAK-WT) by subcellular fractionation and immunoblotting (Fig. 4a, Supplementary Fig. 4a), as we have described previously[25]. In addition, we verified that nuclear targeting-defective FAK, in which the nuclear localisation signal is mutated[25,26] (FAK-NLS), is markedly depleted in enriched nuclei and chromatin extracts of SCC cells expressing FAK-NLS (Fig. 4a, Supplementary Fig. 4b) but is expressed at similar levels as FAK-WT (Fig. 4b). We next interrogated the influence of nuclear FAK on the nuclear proteome. Analysis of isolated nuclear fractions using LC-MS/MS (Fig. 4c) quantified 3,726 proteins in at least three out of four biological replicate experiments (Supplementary Data 5). The nuclear proteomes of SCC FAK−/−, FAK-WT and FAK-NLS cells showed some separation in principal component space (Fig. 4d), and 50 nuclear proteins were differentially regulated according to FAK status ($P < 0.05$, one-way ANOVA with FDR correction; Supplementary Data 6). Proteins involved in cell adhesion and migration, and those that localise to the cell surface or actin cytoskeleton, were over-represented in the set of FAK-dependent nuclear proteins ($P < 0.05$, hypergeometric test with Benjamini–Hochberg correction; Supplementary Fig. 4c, d, Supplementary Data 7). Indeed, of the nuclear proteins differentially regulated in SCC FAK−/− or FAK-NLS cells, half (25 proteins) were cell adhesion proteins, with 36% (18 proteins) annotated as meta-adhesome components (Fig. 4e, Supplementary Data 6). The abundance of almost all proteins dysregulated in the absence of FAK showed a similar trend in the presence of nuclear targeting-defective FAK (Fig. 4f, g), suggesting that nuclear FAK substantially influenced their nuclear abundance.

As it is now well documented that nuclear FAK interacts with transcriptional regulators to control gene expression[25–27], we next addressed whether FAK influenced nuclear protein abundance at the level of mRNA transcription or, if not, via inferred effects on nuclear translocation. We performed RNA-Seq analysis of cells that express or do not express FAK, and we integrated these data, together with previously reported microarray data from the same cells[25], with the FAK-dependent nuclear proteome (Fig. 4g, Supplementary Data 8). Multi-omic analysis revealed that FAK-dependent nuclear protein and mRNA levels clustered together and were positively correlated (FAK-WT median ρ = 0.64, $P < 0.0031$; FAK−/− median ρ = 0.49, $P < 0.047$, Spearman's tests; Fig. 4g–i, Supplementary Fig. 4e, f, Supplementary Data 9), demonstrating that nuclear protein abundance was associated with corresponding mRNA transcription. Furthermore, the abundance of dysregulated nuclear proteins in FAK-NLS cells correlated, and clustered, with corresponding mRNA levels in FAK-deficient (FAK−/−) cells (Fig. 4h, i, Supplementary Fig. 4e, f). For example, platelet-derived growth factor receptor β (PDGFR-β), a cell-surface receptor that can translocate to the nucleus[59], increased in nuclear abundance in the absence of FAK, or when FAK's translocation to the nucleus was impaired, and expression of *Pdgfrb* was concomitantly increased in the absence of FAK expression (Fig. 4e–g, Supplementary Fig. 4f). The majority of adhesion proteins for which nuclear abundance was FAK- or FAK nuclear targeting-dependent were also dysregulated in a similar manner at the level of mRNA transcription (68% with $P < 0.05$; Fig. 4g). These data indicate that nuclear FAK regulates the nuclear abundance of a subset of cell-surface or cytoskeletal adhesion proteins, and this is largely at the level of gene expression rather than nuclear translocation per se. Although we do not investigate the mechanisms by which FAK regulates transcription of these genes here, we have already reported previously that FAK associates with transcriptional complexes in the nucleus to influence gene expression in the case of *Ccl5* and *Il33*, including via chromatin accessibility[25,27,28].

**FAK associates with the adhesome and nuclear protein Hic-5 in the nucleus**. As mentioned, FAK has been reported extensively to generate nuclear molecular complexes that regulate gene expression[25–28]. To determine whether FAK also associates with other adhesion proteins that locate to the nucleus to regulate gene expression, we examined the FAK-proximal nuclear interactome using the proximity-dependent biotinylation technique BioID[60]. We fused FAK to the promiscuous mutated biotin ligase BirA* and expressed this in otherwise FAK-deficient cells (SCC FAK−/− cells) (Fig. 5a). LC-MS/MS analysis of biotinylated proteins affinity purified from nuclear extracts identified 58

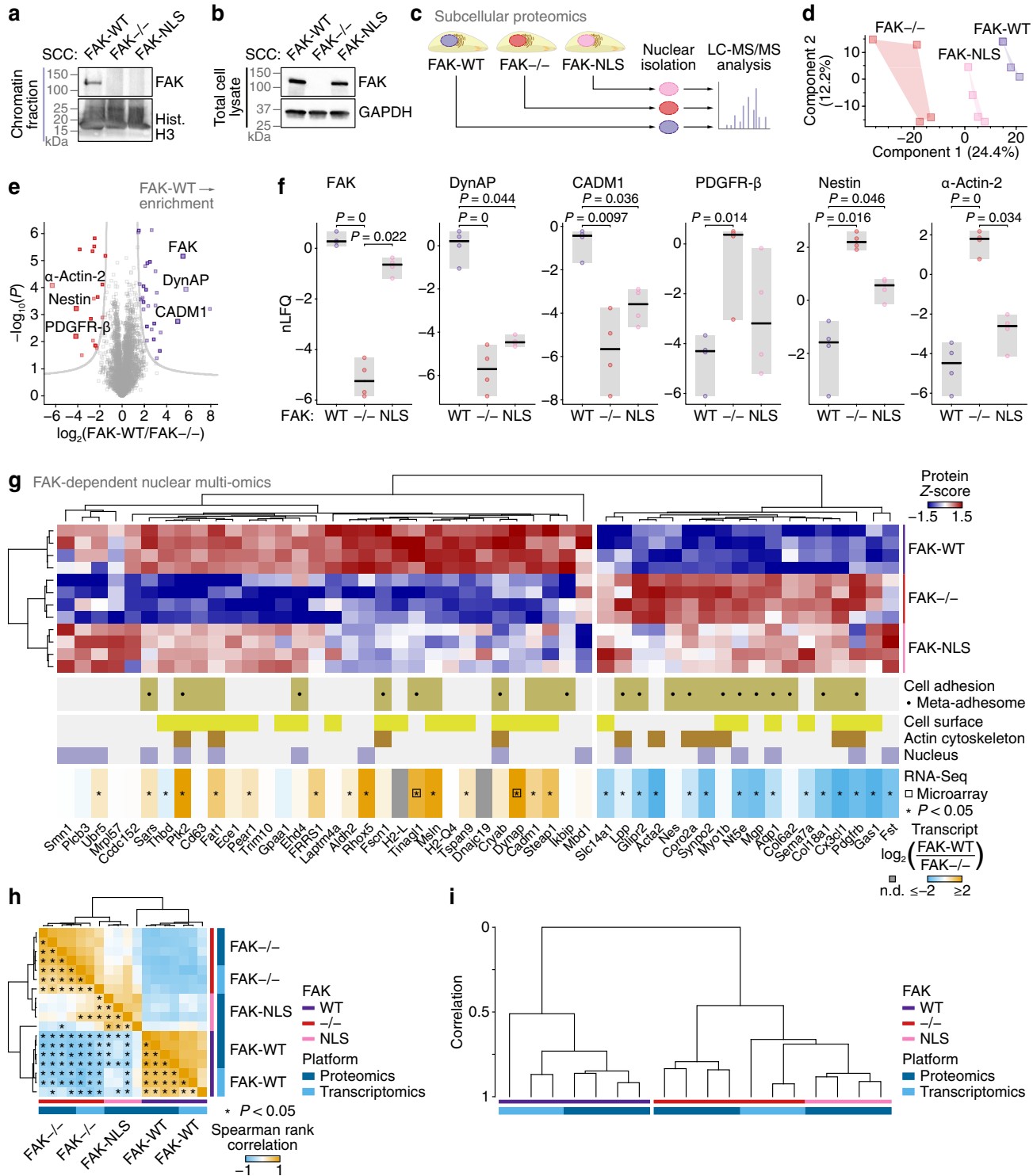

putative FAK-proximal nuclear proteins significantly enriched over control cells expressing unfused BirA* (an empty-vector negative control for nonspecific interactions) ($P < 0.05$, Student's $t$-test with FDR correction) (Supplementary Data 10). Of these nuclear proteins, several were adhesion-associated proteins, with 9 (15.5%) in the consensus adhesome and 11 (19.0%) in the literature-curated adhesome (Fig. 5b, c). Network analysis showed that FAK-proximal nuclear adhesion proteins formed a subnetwork of interacting proteins, which included the key FAK binding partner paxillin, which is also localised to the nucleus[48] (Fig. 2h) and contains LIM domains[18], as well as Hic-5, another paxillin

family member and nuclear-shuttling LIM domain-containing consensus adhesome protein[61,62] (Fig. 5d, Supplementary Fig. 5a). Immunofluorescence confirmed that Hic-5 localises to the nucleus (Fig. 5e), accumulating in the nucleus upon inhibition of exportin-1 using leptomycin B (Fig. 5e, f).

To assess commonalities between FAK- and Hic-5-associated proteins in the nucleus, we examined the Hic-5-proximal nuclear interactome using BioID (Supplementary Fig. 5b). Of the Hic-5-proximal nuclear proteins identified by LC-MS/MS analysis (Supplementary Data 11), a similar number to the FAK-proximal nuclear interactome were adhesion-associated proteins (12

**Fig. 4 Integrative analysis of the FAK-dependent nuclear proteome. a, b** Chromatin extracts (**a**) and total cell lysates (**b**) from SCC cells that express FAK-WT or FAK-NLS or do not express FAK (FAK−/−). Immunoblots are representative of four independent experiments. Hist. H3, histone H3. **c** Workflow for mass spectrometric characterisation of the FAK-dependent nuclear subproteome of SCC cells. **d** Principal component analysis of proteins quantified in at least three out of four biological replicate experiments. **e** Volcano plot of the FAK-dependent nuclear subproteome (FAK-WT versus FAK−/−). Differentially enriched proteins are coloured (purple, enriched in SCC FAK-WT nuclei; red, enriched in SCC FAK−/− nuclei; $P < 0.05$, two-sided Student's $t$-test with FDR correction). Data points representing differentially enriched cell adhesion proteins are indicated by dark shading. Cell adhesion proteins or those associated with the cytoskeleton that were differentially enriched by at least 16 fold are labelled (large data points). **f** Protein abundance in enriched nuclei of SCC FAK−/−, FAK-WT and FAK-NLS cells for proteins labelled in **e**. Black bar, median; light grey box, range. Statistical analysis, two-sided Student's $t$-test with FDR correction ($n = 4$ independent biological replicates). **g** Hierarchical cluster analysis of the FAK-dependent nuclear subproteome. The FAK-dependent nuclear subproteome and FAK-dependent transcriptomes of SCC cells were integrated. Proteins are labelled with gene names for clarity. Open black squares indicate transcripts quantified by microarray analysis. n.d., not determined. *$P < 0.05$; statistical analysis, two-sided Wald test with Benjamini–Hochberg correction ($n = 3$ independent biological replicates); exact $P$ values are provided in Supplementary Data 8. **h** Multi-omic correlation analysis. Spearman rank correlation coefficients for all pairwise sample comparisons in the integrated FAK-dependent nuclear multi-omic data were analysed by hierarchical clustering. *$P < 0.05$; statistical analysis, two-sided Spearman's test; significant correlations are indicated for entries below the main diagonal of the symmetric matrix; exact $P$ values are provided in Supplementary Data 9. **i** Integrative cluster analysis of the multi-omic dataset. The dendrogram scale represents correlation computed as $1 − d$, where $d =$ Euclidean distance. For full cluster analysis, see Supplementary Fig. 4. Source data are provided as a Source Data file.

consensus adhesome and 15 literature-curated adhesome proteins; Supplementary Fig. 5c, d), which formed an interconnected network of proteins (Supplementary Fig. 5e). Consistent with FAK–Hic-5 association, the network of Hic-5-proximal nuclear adhesion proteins included FAK (Fig. 5g, Supplementary Fig. 5e), confirming our FAK nuclear BioID findings (Fig. 5d, Supplementary Fig. 5a). This provided confidence of the reciprocal association between FAK and Hic-5 in the nucleus. Several adhesion proteins (8 proteins) were identified in both nuclear interactomes (53.3% of FAK-proximal adhesion proteins were also Hic-5 proximal; Supplementary Fig. 5f), suggesting that these adhesion protein subnetworks may function together in the nucleus (Fig. 5g, Supplementary Fig. 5g).

Interestingly, we observed that Hic-5 protein expression was upregulated in the absence of FAK, or when FAK's nuclear translocation was impaired (Fig. 5h). As Hic-5 also functions as a transcription co-regulator[63], we investigated a potential co-dependency mechanism between nuclear FAK and Hic-5. We depleted Hic-5 expression using two independent shRNAs (Fig. 5i) and quantified the expression of adhesion-related proteins that we show here are regulated by nuclear FAK (Fig. 4e–g). We used RT-qPCR to verify that gene expression of three of the adhesion- or cytoskeleton-associated proteins most differentially enriched according to FAK status – dynAP, CADM1 and PDGFR-β – was significantly dysregulated in the absence of FAK (Fig. 5j). We further showed that expression of these genes was similarly dysregulated in FAK-NLS cells, in which FAK's nuclear translocation was impaired (Fig. 5j), in accordance with our multi-omics analysis (Fig. 4g), indicating that their transcription is regulated by nuclear FAK. Crucially, we found that expression of *Dynap* and *Cadm1* was downregulated and *Pdgfrb* was upregulated upon depletion of Hic-5 (Fig. 5k), as we also observed in FAK−/− and FAK-NLS cells (Fig. 5j), identifying a co-regulation of certain genes by nuclear FAK and Hic-5. Together, these data imply that nuclear FAK and Hic-5 work together, in a nuclear adhesion protein subcomplex, to co-regulate a subset of genes transcriptionally.

## Discussion

In the context of this description herein of a cancer cell nucleo-adhesome, our work suggests that classical integrin-associated adhesion proteins that translocate to the nucleus, including FAK as we, and others, reported previously[22–28], have important nuclear functions. We show here the exemplar that nuclear FAK, as part of a nuclear adhesion protein network, forms a

compensatory association with Hic-5 to co-regulate the expression of a subset of cytoskeletal and adhesion proteins that can localise to the nucleus; this adds to our understanding of the mechanisms by which nuclear FAK is known to affect transcription and cancer cell communication with the extracellular environment[25–28]. It also indicates the principle that there are complexes between adhesion proteins in the nucleus that function together to regulate transcription.

More generally, our characterisation of a nucleo-adhesome, consisting of more than half of core consensus adhesome proteins, establishes the true extent of adhesion protein translocation to the nucleus, which had not been fully appreciated until now. This work used a cancer cell model but is in keeping with disparate evidence in multiple cell types, mainly using immuno-fluorescence, that adhesion proteins are frequently found in the nucleus, especially under conditions of cellular stress (reviewed in refs. [4,18–21]). We still do not know the extent of the biological consequences of the nuclear presence of most of these adhesion proteins. In addition, the mechanisms mediating the nuclear translocation of adhesion proteins, including adhesion-linked kinases, are not well understood (reviewed in ref. [64]), and it remains to be determined whether, and if so how, adhesion proteins are directly trafficked from adhesion sites to the nucleus or whether they represent completely separate protein pools.

One hypothesis supported by our nucleo-adhesome data is that LIM domain-containing adhesion proteins, such as paxillin and Hic-5 and many others, can shuttle between the cell surface and the nucleus[19]. Proteomic analyses of cell-ECM adhesions have identified that LIM domain-containing proteins are preferentially incorporated into more mature adhesions under myosin II-generated tension[15,30,31], and some LIM domains have been shown to bind mechanically strained actin filaments[65,66]; therefore, we postulate that LIM domain-containing nucleo-adhesome proteins likely constitute a key mechanosensory adhesome module to transduce biomechanical signals from focal adhesions to the nucleus. Indeed, several of these LIM domain-containing proteins have been reported to modulate gene expression (e.g. FHL2 (ref. [67]), TRIP6 (ref. [68])), and we show here that Hic-5 regulates adhesion-related genes, but the mechanisms interconnecting, or partitioning, the cytoplasmic and nuclear functions of the majority of these nucleo-adhesome proteins remain to be established.

Our proteome-wide nucleo-adhesome data report, at a systems level, the remarkable scale of nuclear localisation of adhesion proteins, and we show that nuclear adhesion protein subcomplexes function together to control transcription. These data will serve to

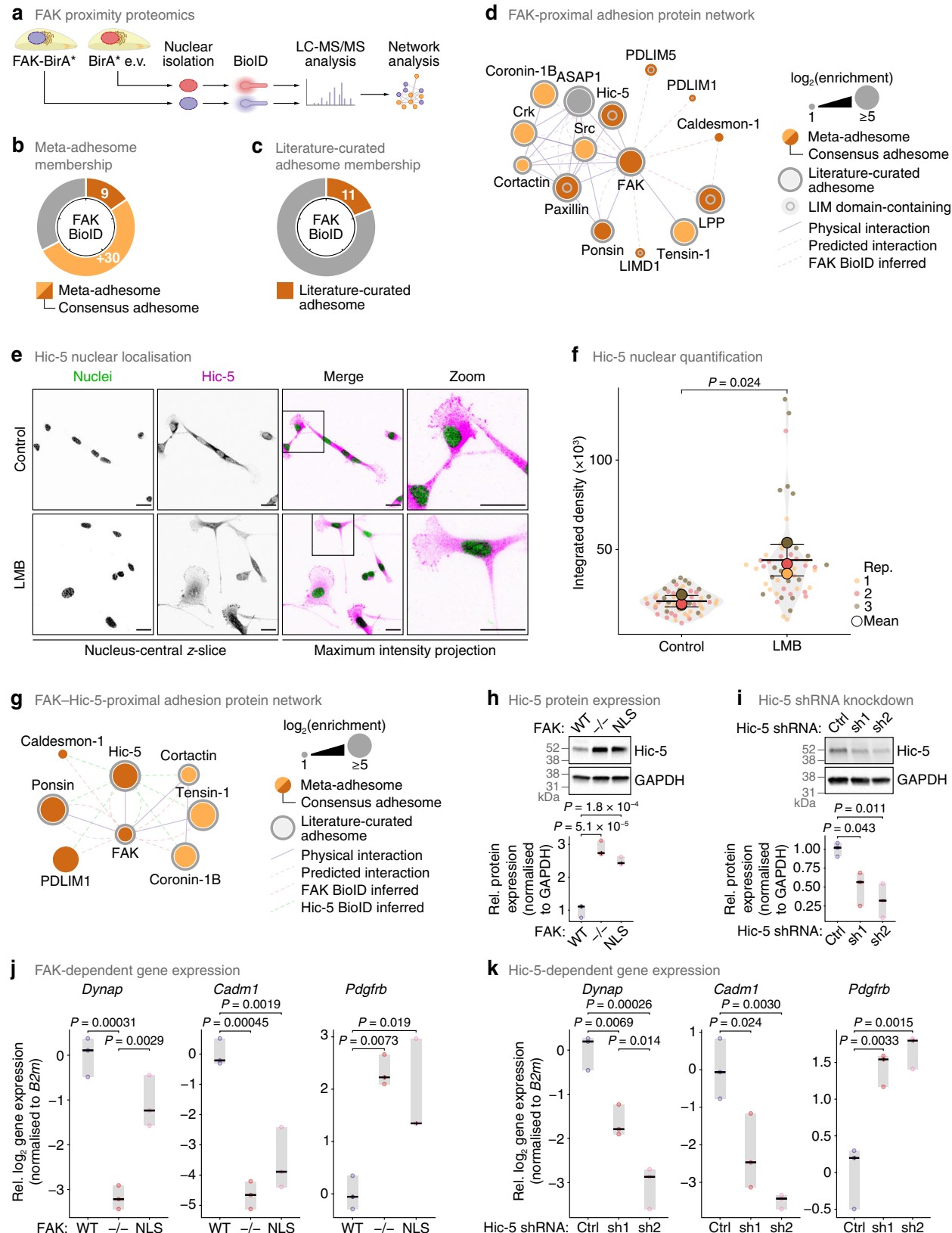

**a** FAK proximity proteomics

**b** Meta-adhesome membership

**c** Literature-curated adhesome membership

**d** FAK-proximal adhesion protein network

**e** Hic-5 nuclear localisation

**f** Hic-5 nuclear quantification

**g** FAK–Hic-5-proximal adhesion protein network

**h** Hic-5 protein expression

**i** Hic-5 shRNA knockdown

**j** FAK-dependent gene expression

**k** Hic-5-dependent gene expression

prime further interrogation of the multiple subcellular functions and shuttling of adhesion proteins.

## Methods

**Cell line generation.** The mouse SCC FAK cellular model and the FAK-NLS mutant were generated in previous studies[25,69]. Generation of the SCC FAK cellular model[69] had University of Glasgow ethical approval and was carried out in accordance with the United Kingdom Animal Scientific Procedures Act (1986) under Home Office Project Licence number 60/4248. Briefly, SCC tumours were induced in K14CreER-*Ptk2*[flox/flox] FVB mice using the 7,12-dimethylbenz[*a*]anthracene–12-*O*-tetradecanoylphorbol-13-acetate two-stage chemical carcinogenesis protocol[69]. SCC cells were established from excised tumours by allowing small tissue pieces to adhere to and grow out on tissue-culture dishes. To induce

**Fig. 5 Analysis of the FAK-proximal nuclear subproteome identifies nuclear interaction with Hic-5. a** Workflow for mass spectrometric characterisation of FAK-proximal nuclear proteins. **b, c** Proportions of specific FAK-proximal proteins ($P < 0.05$, one-sided Student's $t$-test with FDR correction) quantified in the meta-adhesome, including the consensus adhesome (**b**), and the literature-curated adhesome (**c**). Tick marks indicate 20% increments. **d** Interaction network analysis of FAK-proximal nuclear proteins present in the consensus or literature-curated adhesomes, clustered according to connectivity (reported interactions or proximal associations inferred from BioID data; edges). **e** Confocal imaging of SCC cells in the presence or absence of 10 nM LMB, as for Fig. 2h, except magenta is Hic-5. Images are representative of three independent experiments. Scale bars, 20 µm. **f** Quantification of nuclear Hic-5 signal in nucleus-central $z$-slices (see **e**). Black bars, condition mean (thick bar) ± s.d. (thin bars); grey silhouette, probability density. Data from different independent biological replicates (rep.) are indicated by coloured circles; replicate means are indicated by large circles. Statistical analysis, two-sided Welch's $t$-test of replicate means ($n = 54$ and 52 cells for control and LMB treatment, respectively, from $n = 3$ independent biological replicates). **g** Interaction network analysis of the intersection of specific FAK- and Hic-5-proximal nuclear proteins present in the consensus or literature-curated adhesomes, clustered as for **d**. **h** Total cell lysates from SCC cells (top) and quantification of Hic-5 protein expression (bottom). Protein expression was normalised to GAPDH and expressed relative to SCC FAK-WT. **i** Total cell lysates from SCC FAK-WT cells transfected with two independent Hic-5 shRNAs (sh1, sh2) or empty-vector control (Ctrl) (top) and quantification of Hic-5 protein expression (bottom, as for **h**). **j, k** RT-qPCR analyses of expression of selected genes identified by nuclear multi-omic analysis (see Fig. 4) in SCC cells (**j**) and cells depleted of Hic-5 (**k**). Gene expression was normalised to *B2m*, binary-logarithm transformed and expressed relative to FAK-WT (**j**) or Ctrl (**k**) cells. For **h**–**k**, black bar, median; light grey box, range. Statistical analysis, one-way ANOVA with Tukey's correction ($n = 3$ independent biological replicates). Source data are provided as a Source Data file.

*Ptk2* deletion, SCC cells were treated with 15 µM 4-hydroxytamoxifen for 24 h and single cells seeded into each well of a 96-well plate to establish colonies. Colonies were screened for loss of *Ptk2* by PCR genotyping and verified by immunoblotting to identify FAK−/− cells. To generate the FAK-NLS mutant, PCR-based site-directed mutagenesis of FAK-WT was used to introduce alanine point mutations at arginine-177, arginine-178, lysine-190, lysine-191, lysine-216 and lysine-218. Mutational status was confirmed using sequencing. Retroviral transduction was used to stably re-express FAK-WT or the FAK-NLS mutant in FAK−/− cells, using two rounds of infection per cell line[25], which were selected with 0.25 mg/ml hygromycin.

For BioID, *Ptk2* (forward primer, 5′-ACCTTGATCCAAAC-3′; reverse primer, 5′-GCTGATCATTTTCAGTC-3′) or *Tgfb1i1* (forward primer, 5′-GCGGCCGCA TGGAGGACCTGGATGCCCTGC-3′; reverse primer, 5′-GGATCCTCAGCCG AAGAGCTTCAGGAAAGC-3′) were amplified by PCR and cloned into the pQCXIN-BirA\*-Myc retroviral vector. Vector containing *Ptk2* (FAK-BirA\*) or BirA\* empty vector was introduced into SCC FAK−/− cells; vector containing *Tgfb1i1* (Hic-5-BirA\*) or BirA\* empty vector was introduced into SCC FAK-WT cells. Cells were infected, selected and expanded as previously described[27]. Briefly, Phoenix Ecotropic cells were transfected with BirA\* vector or empty vector using Lipofectamine 2000 (Thermo Fisher Scientific) following the manufacturer's instructions. Cell culture supernatant was removed after 48 h, filtered (0.45-µm Millex-HA filter; Merck Millipore), diluted in cell growth medium (see below), supplemented with 5 µg/ml polybrene and added to cells for 24 h. Cells were subjected to two rounds of infection and then selected with 400 µg/ml G418.

For knockdown of Hic-5, pLKO.1 lentiviral vector containing mouse *Tgfb1i1* shRNA (the RNAi Consortium clone identifiers TRCN0000075518 (sh1) or TRCN0000075519 (sh2), Horizon Discovery) or empty vector was introduced into SCC FAK-WT cells using standard lentivirus procedures[27]. Cells were subjected to two rounds of infection and then selected with 2 µg/ml puromycin.

**Cell culture**. Mouse SCC cells were grown at 37 °C, 5% (v/v) CO₂, in Glasgow minimum essential medium (MEM) (Sigma-Aldrich) supplemented with 10% (v/v) foetal bovine serum, 1× MEM nonessential amino acids (both Life Technologies), 2 mM L-glutamine, 1 mM sodium pyruvate and 1× MEM vitamins (all Sigma-Aldrich) (complete medium). Cell lines were routinely tested for mycoplasma and used within three months of recovery from frozen.

**Subcellular fractionation**. For nuclear enrichment, we modified an isotonic buffer-mediated cellular dissection method[36]. Cells were seeded in 150 mm-diameter dishes ($4 \times 10^6$ cells/dish) and grown for 48 h. Cells were washed with ice-cold phosphate-buffered saline (PBS) and lysed with buffer A (20 mM Tris-HCl, pH 7.5, 1 mM MgCl₂, 1 mM EGTA, 0.03% (w/v) NP-40), supplemented with protease and phosphatase inhibitor cocktails, for 5 min at 4 °C under gentle rotation. Lysates were fractionated by centrifugation at $800 \times g$ for 4 min at 4 °C to pellet nuclear and perinuclear material. To the supernatant, NP-40 was added to a final concentration of 1% (w/v) and SDS to 0.1% (w/v), after which the lysate was clarified by centrifugation ($16,000 \times g$ for 15 min at 4 °C) and the clarified cytoplasmic fraction was retained. The crude nuclear/perinuclear pellet was washed once with buffer A and once with buffer A without NP-40 and gently resuspended in two pellet-volumes of buffer B (10 mM Tris-HCl, pH 7.5, 2.5 mM MgCl₂, 1.5 mM KCl, 0.2 M LiCl, 0.1% (w/v) sodium deoxycholate, 0.1% (w/v) Triton X-100), supplemented with protease and phosphatase inhibitor cocktails, for 15 min at 4 °C under gentle rotation. Lysates were fractionated by centrifugation at $2000 \times g$ for 5 min at 4 °C to pellet nuclear material. To the supernatant, SDS was added to a final concentration of 0.1% (w/v), after which the lysate was clarified by centrifugation ($16,000 \times g$ for 15 min at 4 °C) and the clarified perinuclear fraction was retained. The pellet was washed once with buffer B and once with buffer B without sodium deoxycholate or Triton X-100 and resuspended in two pellet-volumes of RIPA lysis buffer (50 mM Tris-HCl, pH 7.4, 150 mM NaCl, 5 mM EGTA, 0.1% (w/v) SDS, 0.5% (w/v) sodium deoxycholate, 1% (w/v) NP-40), supplemented with protease and phosphatase inhibitor cocktails. Lysates were sonicated (5× cycles of 30 s on and 30 s off; Bioruptor, Diagenode) and clarified by centrifugation ($16,000 \times g$ for 15 min at 4 °C) and the clarified nuclear fraction was retained. Isolated fractions were analysed by immunoblotting or processed for LC-MS/MS analysis (see below).

**Chromatin isolation**. Chromatin preparation was performed as previously described[27,70]. Briefly, cells were washed twice with ice-cold PBS and lysed with chromatin extraction buffer (10 mM Hepes, pH 7.9, 10 mM KCl, 1.5 mM MgCl₂, 0.34 M sucrose, 10% (w/v) glycerol, 0.2% (w/v) NP-40), supplemented with protease and phosphatase inhibitor cocktails. Lysates were fractionated by centrifugation at $6,500 \times g$ for 5 min at 4 °C to pellet nuclear material. The nuclear pellet was washed with chromatin extraction buffer without NP-40 and centrifuged at $6,500 \times g$ for 5 min at 4 °C. The pellet was gently resuspended in low-salt buffer (10 mM Hepes, pH 7.9, 3 mM EDTA, 0.2 mM EGTA), supplemented with protease and phosphatase inhibitor cocktails, incubated for 30 min at 4 °C under gentle rotation and centrifuged at $6,500 \times g$ for 5 min at 4 °C. The pellet was gently resuspended in high-salt solubilisation buffer (50 mM Tris-HCl, pH 8.0, 2.5 M NaCl, 0.05% (w/v) NP-40), supplemented with protease and phosphatase inhibitor cocktails, vortexed briefly and incubated for 30 min at 4 °C under gentle rotation. Lysates were fractionated by centrifugation at $6,500 \times g$ for 5 min at 4 °C to pellet nonchromatin material. The chromatin-containing supernatant was clarified by centrifugation ($16,000 \times g$ for 10 min at 4 °C) and the clarified chromatin fraction was retained. Proteins in the clarified chromatin fraction were precipitated by addition of ice-cold trichloroacetic acid to 10% (v/v) final concentration and incubation for 15 min at 4 °C. Samples were centrifuged at $21,000 \times g$ for 15 min at 4 °C, and the pellet was washed twice with ice-cold acetone (500 µl). Samples were clarified by centrifugation ($21,000 \times g$ for 15 min at 4 °C), and each pellet was collected and air-dried. Chromatin extracts were analysed by immunoblotting (see below).

**Immunoblotting**. For total cell lysates, cells were washed twice with ice-cold PBS and lysed with RIPA lysis buffer, supplemented with cOmplete ULTRA protease inhibitor and PhosSTOP phosphatase inhibitor cocktails (Sigma-Aldrich). Total cell lysates or subcellular fractions were clarified by high-speed centrifugation ($16,000 \times g$ for 15 min at 4 °C). Protein concentration was estimated by BCA protein assay (Thermo Fisher Scientific), and 15 µg of total protein were supplemented with 2× SDS sample buffer (100 mM Tris-HCl, pH 6.8, 20% (w/v) glycerol, 4% (w/v) SDS, 10% (v/v) β-mercaptoethanol, bromophenol blue) and incubated for 5 min at 95 °C. Samples were resolved by polyacrylamide gel electrophoresis using 4–15% Mini-PROTEAN TGX gels (Bio-Rad), and proteins were transferred to nitrocellulose or polyvinylidene fluoride membrane and blocked with 5% (w/v) bovine serum albumin in Tris-buffered saline containing 0.1% (w/v) Tween 20 (BSA/TBS-T). Membranes were probed with the following primary antibodies (all Cell Signalling Technology, diluted 1:1000 in BSA/TBS-T, unless otherwise stated): α-tubulin (#3873, diluted 1:2000), glyceraldehyde-3-phosphate dehydrogenase (GAPDH) (#2118; #5174), protein disulphide isomerase A4 (ERp72) (#5033), receptor-binding cancer antigen expressed on SiSo cells (RCAS1) (#12290), golgin subfamily A member 2 (GM130) (#610822, BD Biosciences), SUN domain-containing protein 2 (Sun2) (#ab87036, Abcam), lamin-A/C (#4777), poly(ADP-ribose) polymerase 1 (PARP1) (#9532), histone H4 (#2935), AFAP1 (#610200, BD Biosciences), c-Met (#4560), APPL1 (#3858), EEA1 (#2411), cathepsin B (#31718), golgin-97 (#13192), testin (#sc-373913, Santa Cruz Biotechnology), FAK (#3285), histone H3 (#4620), HP1α/β (#2623), Hic-5 (#611164, BD Biosciences). Bound antibodies were detected by incubation with anti-rabbit, anti-mouse or streptavidin

HRP-conjugated secondary antibodies (Cell Signaling Technology) and were visualised using a ChemiDoc MP Imaging System (Bio-Rad) and analysed using Image Lab (version 5.2.1). Uncropped scans are provided as a Source Data file.

**Immunofluorescence and confocal microscopy**. Cells were seeded on glass coverslips ($2 \times 10^4$ cells/coverslip) and grown for 16 h. Cells were treated with 10 nM leptomycin B or vehicle control (ethanol) for 4 h or with 1 mM hydrogen peroxide or vehicle control (water) for 1 h. Cells were washed once with TBS, permeabilised and fixed with fixation buffer (3.7% (v/v) formaldehyde, 100 mM PIPES, 10 mM EGTA, 1 mM MgCl$_2$, 0.2% (w/v) Triton X-100) and incubated for 10 min at room temperature. Cells were washed twice with wash buffer (0.1% (w/v) Triton X-100 in TBS) and blocked with block buffer (2% (w/v) BSA in TBS containing 0.1% (w/v) Triton X-100) for 30 min at room temperature. Coverslips were stained with the following primary antibodies (all diluted 1:200 in block buffer, unless otherwise stated): Hic-5 (#611164, diluted 1:150), paxillin (#612405, BD Biosciences), testin (#sc-271184, Santa Cruz Biotechnology). Cells were washed three times with wash buffer, and bound antibodies were detected by incubation with anti-mouse Alexa Fluor 488-conjugated secondary antibody (#A11001, Invitrogen) diluted 1:400 in block buffer. Coverslips were washed three times with wash buffer and mounted in ProLong Glass antifade mountant with NucBlue stain (Invitrogen).

Images were acquired on an FV3000 confocal microscope (Olympus) using a UPlanSApo 40× 0.95-NA or a UPlanSApo 60× 1.35-NA objective. Data were collected with a scan format of 1024 × 1024 pixels and pixel dwell of 4 μs. Confocal images were acquired with z-step size of 0.5 μm. Images were analysed using Fiji (version 1.53 h)[71].

**Subcellular proteome analysis**. Proteins (200 μg) from subcellular fractions isolated using the modified isotonic buffer-mediated cellular dissection method (detailed above) were precipitated by addition of 5 volumes of ice-cold acetone and incubation for 2 h at −20 °C. Samples were centrifuged at 16,000 × g for 10 min at 4 °C and the supernatant was removed with a glass Pasteur pipette. Ice-cold acetone (500 μl) was added, and samples were incubated for 16 h at −20 °C. Samples were sonicated (10× cycles of 30 s on and 30 s off; Bioruptor) and clarified by centrifugation (16,000 × g for 10 min at 4 °C). Pellets were washed with ice-cold acetone (500 μl) and incubated for 1 h at −20 °C. Samples were clarified by centrifugation (16,000 × g for 10 min at 4 °C), and each pellet was collected and air-dried.

Protein pellets were resuspended in 8 M urea, 200 mM Tris-HCl, pH 8.9, and sonicated (5× cycles of 30 s on and 30 s off; Bioruptor). Proteins (40 μg) were reduced with 10 mM dithiothreitol for 30 min at 37 °C and then alkylated with 25 mM iodoacetamide for 30 min at room temperature in the dark. To samples, 200 mM Tris-HCl, pH 8.9, 10 mM DTT was added to dilute urea concentration from 8 M to 6 M, and samples were incubated with MS-grade Lys-C (Alpha Laboratories) (1:50 enzyme:protein ratio) for 3–4 h at 37 °C. Samples were further diluted from 6 M to 2 M urea concentration, and samples were incubated with sequencing-grade trypsin (Promega) (1:50 enzyme:protein ratio) for 16 h at 37 °C. Peptides were acidified with trifluoroacetic acid (~1% (v/v) final concentration), desalted on homemade C18 StageTips and resuspended in 0.1% (v/v) trifluoroacetic acid. Purified peptides were analysed by LC-MS/MS (see below).

**Proximal interactome analysis by BioID**. Cells were incubated with 50 μM biotin (Merck Millipore) for 16 h at 37 °C, washed twice with ice-cold PBS and lysed with cyto buffer (10 mM Tris-HCl, pH 7.5, 3 mM MgCl$_2$, 1 mM EGTA, 0.05% (w/v) NP-40), supplemented with protease and phosphatase inhibitor cocktails, for 5 min at 4 °C. Lysates were fractionated by centrifugation at 1000 × g for 5 min at 4 °C to pellet nuclear material. The pellet was washed with cyto buffer and resuspended in RIPA lysis buffer, supplemented with protease and phosphatase inhibitor cocktails. Lysates were sonicated (3× cycles of 1 min on and 2 min off; Bioruptor) and clarified by centrifugation (16,000 × g for 15 min at 4 °C) and the clarified nuclear fraction was retained. Protein concentration was estimated by BCA protein assay, and 2 mg of total protein was incubated with streptavidin-conjugated magnetic beads (Dynabeads MyOne Streptavidin C1, Thermo Fisher Scientific) for 16 h at 4 °C. Beads were washed using a magnetic tube rack three times with ice-cold RIPA lysis buffer and twice with ice-cold PBS. Captured proteins were subjected to on-bead proteolytic digestion as previously described[72]. Briefly, proteins were incubated with digestion buffer (0.3 μg trypsin in 2 M urea, 50 mM Tris-HCl, pH 8) for 30 min at 27 °C and then digestion buffer supplemented with 10 mM dithiothreitol for 16 h at 37 °C. Peptides were alkylated with iodoacetamide (55 mM final concentration; 30 min at room temperature), acidified with trifluoroacetic acid (~1% (v/v) final concentration), desalted on homemade C18 StageTips and resuspended in 0.1% (v/v) trifluoroacetic acid. Purified peptides were analysed by LC-MS/MS (see below).

**MS data acquisition**. LC-MS/MS was performed using an UltiMate 3000 RSLCnano system coupled online to a Q Exactive Plus Hybrid Quadrupole-Orbitrap mass spectrometer (QE+) or an Orbitrap Fusion Lumos Tribrid mass spectrometer (Lumos) (all Thermo Fisher Scientific). Peptides were injected onto a C18-packed emitter in buffer A (2% (v/v) acetonitrile, 0.5% (v/v) acetic acid) and eluted with a linear 120-min gradient of 2%–45% (v/v) buffer B (80% (v/v)

acetonitrile, 0.5% (v/v) acetic acid) (for subcellular proteome experiments) or a linear 40-min gradient of 2%–35% (v/v) buffer B (for BioID experiments). Eluting peptides were ionised in positive ion mode before data-dependent analysis. The target value for full scan MS spectra was $3 \times 10^6$ charges in the 300–1,650 $m/z$ range, with a resolution of 70,000 (QE+), or $5 \times 10^5$ charges in the 350–1400 $m/z$ range, with a resolution of 120,000 (Lumos). Ions were fragmented with normalised collision energy of 35 (Lumos) or 26 (QE+; 28 for BioID experiments), selecting the top 12 ions (QE+) or using a 2-s cycle time (Lumos). A dynamic exclusion window of 30 s (for subcellular proteome experiments) or 10 s (for BioID experiments) was enabled to avoid repeated sequencing of identical peptides. The target value for MS/MS spectra was $5 \times 10^4$ ions, with a resolution of 17,500 (QE +), or $1 \times 10^4$ ions (Lumos). All spectra were acquired with 1 microscan and without lockmass.

**MS data analysis**. Label-free quantitative analysis of MS data was performed using MaxQuant (version 1.6.2.10)[73]. Peptide lists were searched against the mouse UniProtKB database (version 2018_07) and a common contaminants database using the Andromeda search engine[74]. For subcellular fractionation profiling of different subcellular fractions, raw files were organised into separate parameter groups according to the subcellular fraction analysed. For all experiments, cysteine carbamidomethylation was set as a fixed modification; methionine oxidation, N-terminal glutamine cyclisation, N-terminal carbamylation (except for BioID experiments), biotin (BioID experiments only) and protein N-terminal acetylation were set as variable modifications (up to five modifications per peptide). Peptide identifications in one or more LC runs that were not identified in other LC runs were matched and transferred between runs (time window of 0.7 min for QE+, 1.0 min for Lumos). MS/MS were required for quantitative comparisons (except for subcellular fractionation profiling of different subcellular fractions), and large label-free quantification (LFQ) ratios were stabilised.

Peptide and protein FDRs were set to 1%, determined by applying a target-decoy search strategy using MaxQuant. Enzyme specificity was set as C-terminal to arginine and lysine, except when followed by proline, and a maximum of two missed cleavages were allowed in the database search. Minimum peptide length was seven amino acids, and at least one peptide ratio was required for LFQ. Proteins matching to the reversed or common contaminants databases and matches only identified by site were omitted.

For subcellular fractionation profiling, to compare different subcellular fractions, LFQ intensities for proteins quantified in four or more biological replicates in at least one experimental group were weighted according to protein yields of corresponding fractions (cytoplasmic, perinuclear/organellar, nuclear) to account for different protein amounts in different subcellular fractions. Intensities were binary-logarithm transformed. Values missing from all biological replicates of an experimental group were imputed from a uniform distribution using the R package imp4p (version 1.0)[75]; remaining missing values were imputed by k-nearest neighbour averaging using the R package impute (version 1.64.0). Components of the consensus adhesome[15] quantified in all five biological replicate nuclear fractions were classified as core nucleo-adhesome proteins.

For nuclear subproteome analysis, to compare nuclear fractions from different cell lines, LFQ intensities for proteins quantified in three or more biological replicates in at least one experimental group were binary-logarithm transformed and sample-median subtracted. Missing values were imputed from a width-compressed, down-shifted Gaussian distribution using Perseus (version 1.5.2.6)[76].

For BioID experiments, LFQ intensities for proteins quantified in all three biological replicates in at least one experimental group were binary-logarithm transformed. BioID intensities were normalised to streptavidin intensities. Missing values were imputed from a width-compressed, down-shifted Gaussian distribution using Perseus.

Statistical significance of differentially enriched proteins was determined by one-way ANOVA and two-sided Student's t-tests (one-sided Student's t-tests for BioID experiments) with artificial within-groups variance set to 1 and a permutation-based FDR threshold of 5% (computing 1000 randomisations). Area-proportional Euler diagrams were computed using parallel particle swarm optimisation (maximum of 1000 iterations) implemented in VennMaster (version 0.38.2)[77].

**Functional enrichment and network analyses**. Proteins were classified as cell adhesion proteins if they were annotated with Gene Ontology terms GO:0007155, GO:0005925, GO:0007160, GO:0007159, GO:0030054 or GO:0005911 or a meta-adhesome component[15]; cell-surface proteins were those annotated with terms GO:0005886, GO:0009986 or GO:0005887; actin-cytoskeletal proteins were those annotated with terms GO:0015629, GO:0007015, GO:0032956, GO:0032432, GO:0032233, GO:0003779, GO:0005884 or GO:0001725; nuclear proteins were those annotated with terms GO:0005634 or GO:0005654. Gene Ontology term associations were determined using AmiGO 2 (version 2.5.12)[78–80]. UniProt sub-cellular location annotation was obtained from the UniProtKB database (version 2021_02)[81]. Cell Atlas subcellular annotation was obtained from the Human Protein Atlas (version 20.1) (https://www.proteinatlas.org)[37]. Over-representation analyses were performed using WebGestalt (version 2019)[82], ToppGene (version 2019-Oct-08 21:31)[83] and g:Profiler (versions e98_eg45_p14_ce5b097 and e101_eg48_p14_baf17f0)[84]. To reduce redundancy of enriched functional

categories, where stated, gene sets were clustered according to Jaccard index and classified with representative terms using affinity propagation via R package APCluster implemented in WebGestalt[82,85,86].

Composite functional association networks were constructed using GeneMANIA (versions 3.5.1 and 3.5.2; human interactions)[87] in Cytoscape (version 3.7.1)[88]. Graph-based analysis of gene-set overlap of enriched functional categories (category membership, 5–5000) was performed using EnrichmentMap (version 3.2.1)[89] in Cytoscape. Subgraphs of gene sets (nodes) with at least five functional categories per connected component were weighted by the Jaccard coefficient of gene-set membership overlap (edges) and clustered using the force-directed algorithm in the Prefuse toolkit[90].

**RNA-Seq analysis**. RNA was extracted from SCC FAK-WT and FAK–/– cells using an RNeasy kit (Qiagen) following the manufacturer's instructions, using 350 μl buffer RLT with 0.1% (v/v) β-mercaptoethanol for the initial step of the protocol. To verify sample quality, RNA samples were assessed using the Bioanalyzer RNA 6000 pico assay (Agilent) run on a 2100 Bioanalyzer instrument (Agilent). All samples had an RNA integrity number (RIN) of at least 9.6 (RIN ≥ 8 considered a suitable quality for sequencing) (Supplementary Table 1). Samples were prepared using the TruSeq RNA Library Prep Kit v2 (low-sample protocol) (Illumina), validated using the High Sensitivity DNA assay (Agilent) run on a 2100 Bioanalyzer instrument (Illumina) and paired-end sequenced using a HiSeq 4000 platform (Illumina) at BGI (Supplementary Table 1, Supplementary Data 8).

Alignment to the *Mus musculus* GRCm38 reference transcriptome was performed using the pseudoalignment software kallisto (version 0.43.1)[91] applying default parameters. Transcript abundance was summarised to gene level (Ensembl-based annotation package EnsDb.Mmusculus.v79) and imported into the differential expression analysis R package DESeq2 (version 1.24.0)[92] using the R package tximport (version 1.2.0)[93]. Genes with zero read counts were removed prior to differential expression analysis. Hypothesis testing was performed in DESeq2 using a two-sided Wald test with Benjamini–Hochberg correction.

**RT-qPCR**. RNA was extracted from cells using an RNeasy kit (Qiagen) following the manufacturer's instructions. RNA (3 μg) was converted to cDNA using a SuperScript II cDNA synthesis kit (Thermo Fisher Scientific). cDNA (62.5 ng) was added to SYBR Green master mix (Thermo Fisher Scientific) with 0.25 μl of 10 μM primers, and the final reaction mixtures were made up to 10 μl with deionised water. Reactions consisted of 3 min at 95 °C, followed by 40 cycles of 15 s at 95 °C, 30 s at 60 °C and 10 s at 72 °C (melt curve analysis, 15 s at 95 °C, 1 min at 60 °C and 15 s at 95 °C). Expression of *Dynap* (forward primer, 5′-CATGTGACATGTG AGCTGCA-3′; reverse primer, 5′-GATGACGTAGAGGGAGAGGC-3′), *Cadm1* (forward primer, 5′-CAACATGCCGTACTGTCTGG-3′; reverse primer, 5′-TGGTAGTGGTGGTGGTTGTT-3′) and *Pdgfrb* (forward primer, 5′-GGAG GTGACGCTACATGAGA-3′; reverse primer, 5′-CCTGGAGGCTGTAGACG TAG-3′) was normalised to *B2m* (forward primer, 5′-CTGCTACGTAACACAGT TCCACC-3′; reverse primer, 5′-CATGATGCTTGATCACATGTCTCG-3′). All primers were optimised for optimal primer efficiencies, and a melt curve analysis was performed for every qPCR experiment performed.

**Dimensionality reduction**. Principal component analysis was performed using Perseus or R. t-distributed stochastic neighbour embedding (t-SNE) of subcellular fractionation profiling data (Supplementary Data 2) was performed using the R packages MSnbase (version 2.15.7)[94] and pRoloc (version 1.30.0)[95], with the perplexity parameter tuned to 50. To generate a curated subcellular marker protein set for dataset annotation, the validated mouse marker protein set from pRoloc was filtered to exclude (i) non-nuclear marker proteins for which there was evidence for nuclear localisation in Gene Ontology, UniProt or Cell Atlas, (ii) nuclear marker proteins for which there was evidence for non-nuclear localisation in Gene Ontology, UniProt or Cell Atlas and (iii) marker proteins that did not have supporting Cell Atlas evidence with supported or enhanced reliability[37]. This curation of marker assignment resulted in a set of 272 core organelle marker proteins that are resident in one of 13 subcellular locations: cytosol, actin cytoskeleton, plasma membrane, endosome, lysosome, mitochondrion, ribosome, peroxisome, proteasome, Golgi, endoplasmic reticulum, chromatin and nucleus (Supplementary Data 12).

**Unsupervised learning**. Binary, agglomerative hierarchical cluster analyses were performed using Cluster 3.0 (C Clustering Library, version 1.54)[96]. For subcellular fractionation profiling, to enable relative comparison of protein abundance in different subcellular fractions, binary-logarithm-transformed LFQ intensities were min-max scaled row-wise (protein-wise) and expressed as scaled intensities in the range [0,1]. For integrative cluster analysis, multiple matrices of Z-transformed normalised abundance derived from proteomic and transcriptomic platforms were concatenated to form a single multi-omic matrix with equally relative-weighted features, selecting only features quantified using both platforms. Spearman rank correlation coefficients or Euclidean distances were calculated and adapted as distances, if necessary, and distance matrices were computed using complete linkage or average linkage, respectively. Hierarchical clustering results were visualised using Java TreeView (version 1.1.5r2)[97].

**Supervised learning**. Classification of subcellular localisation was performed using the SVM algorithm with a radial basis function kernel implemented in pRoloc[95]. The labelled training data were derived from the curated subcellular marker protein set used for dataset annotation (see above), which was consolidated by merging the subcellular marker classes into three marker classes – cytoplasmic, perinuclear/organellar/other and nuclear – to reflect the resolution of the subcellular fractionation procedure and to remove marker classes with insufficient marker proteins for supervised learning (Supplementary Data 12). Consensus adhesome proteins were omitted from the training data. We optimised the free parameters of the SVM using a grid search, applying five-fold cross-validation repeated 100 times, which identified the maximum harmonic mean of precision and recall when cost of constraints violation (misclassification penalty) was 16 and sigma (the inverse kernel width for the radial basis kernel) was 0.1. For generation of the SVM model, classes were weighted inversely to class size to account for class size differences. Scores based on SVM classification of proteins were set an FDR threshold of 10% based on concordance with evidence for subcellular localisation in Gene Ontology, UniProt, Cell Atlas or the literature. Subcellular localisation prediction based on protein sequence information was performed using the deep-learning neural network model implemented in DeepLoc (version 1.0)[98].

**Statistics and reproducibility of experiments**. Distributions of sampled populations were tested for normality using the Shapiro–Wilk test. Statistical significance of image quantification data was calculated using a two-sided Welch's t-test of means from independent biological replicates. Immunoblotting data were analysed using one-way ANOVA with Tukey's correction. For proteomic data analyses, proteins quantified in at least four out of five, at least three out of four or three out of three independent biological replicates for at least one experimental condition were further analysed, and significantly differentially abundant proteins were determined using one-way ANOVA and two-sided Student's t-tests (one-sided Student's t-tests for BioID experiments) with permutation-based FDR correction (computing 1000 randomisations). Differentially transcribed genes derived from RNA-Seq data were determined using a two-sided Wald test with Benjamini–Hochberg correction; RT-qPCR data were analysed using one-way ANOVA with Tukey's correction. For functional enrichment analyses, significantly enriched terms were determined using a one-sided hypergeometric test with Benjamini–Hochberg correction. No statistical method was used to predetermine sample size. Data plots were generated using Excel (Microsoft), Perseus, PlotsOfData (version 1.0.5)[99], Prism (version 8.3.0) (GraphPad), R or SuperPlotsOfData (version 1.0.3)[100].

**Reporting summary**. Further information on research design is available in the Nature Research Reporting Summary linked to this article.

## Data availability

MS data that support the findings of this study have been deposited in ProteomeXchange via the PRIDE partner repository[101] with the dataset accession identifiers PXD025870 (subcellular fractionation profiling), PXD020179 (FAK-dependent nuclear subproteome), PXD025861 (FAK-proximal nuclear interactome) and PXD025868 (Hic-5-proximal nuclear interactome). RNA-Seq data that support the findings of this study have been deposited in the Gene Expression Omnibus[102] with GEO series accession identifier GSE147670. Protein sequence and functional annotation data used in this study are available in the UniProtKB database[81] version 2018_07 (https://ftp.uniprot.org/pub/databases/uniprot/previous_releases/release-2018_07/) and version 2021_02 (https://ftp.uniprot.org/pub/databases/uniprot/previous_releases/release-2021_02/), respectively; protein subcellular localisation data used in this study are available in the Human Protein Atlas[37] version 20.1 (https://v20.proteinatlas.org). All other data supporting the findings of this study are available within the paper and its Supplementary Information and Source Data files. Source data are provided with this paper.

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

## Acknowledgements

We thank Alfonso Bolado, Niall Quinn and Jimi Wills for assistance with MS data acquisition, Martin Lee for assistance with microscopy and Douglas Armstrong, Didier Devaux, Noor Gammoh, Frederic Li Mow Chee, Roza Ali Masalmeh, Christina Schoenherr, Oksana Sorokina and Athanasia Yiapanas for discussions. The work was funded by Cancer Research UK (grants C157/A15703 and C157/A24837 to M.C.F.) and supported by the Wellcome Trust (multiuser equipment grant 208402/Z/17/Z to A.v.K.).

## Author contributions

A.B. and M.C.F. conceived and coordinated the project; A.B., B.G.C.G., A.H., J.C.D., B.S., A.v.K., V.G.B. and M.C.F. designed the experiments and interpreted the results; A.B., B.G.C.G., A.H., A.E.P.L., E.S.K., N.M. and J.C. performed the experiments; A.B., B.G.C.G., A.H., A.E.P.L., E.S.K., L.K. and G.R.G. analysed the data; A.B. and M.C.F. wrote the paper, with contributions from B.G.C.G. and A.H.; all authors reviewed and approved the final version of the manuscript.

## Competing interests

The authors declare no competing interests.
