## [Peer Review File · Nature Communications]

Reviewers' comments:

Reviewer #1 (Remarks to the Author):

This manuscript focuses round the study of a subset of proteins involved in the formation of cell adhesion proteins that exist also within the nucleus in an attempt to understand potential moonlighting and other non-canonical roles of these proteins. The authors first determine the likely subset of nuclear localized adhesion proteins using informatics approaches on existing datasets, and create a list of adhesome proteins likely in associate with the nucleus.

The authors then use a detergent based subcellular fraction method to enrich for nuclear, perinuclear and cytoplasmic proteins in a squamous cell carcinoma (SCC) cell line and carry out quantitative mass spectrometry to determine enrichment of adhesome and associated proteins within each fraction and identify some of the above adhesome proteins as present in the nuclear fraction. Using a stringent list of these proteins they construct a network and describe sets of proteins with hitherto ascribed nuclear location from known adhesome proteins.

The authors then concentrate on two kinases, FAK and ILK, that are known to associate with the adhesome and attempt to determine the nuclear function of these proteins. In the case of FAK they assign a function in expression of a subset of cell surface or cytoskeletal proteins. Using a proximity tagging and immune-precipitation strategies, they determine interacting partners of ILK and find that a substantial proportion of interacting partners have the potential to locate within the nucleus. Knock-outs of ILK did not affect the nuclear proteome but the authors were able to demonstrate that cell survival was impaired in ILK knock-out cell lines.

In conclusion, the authors postulate that 70% of core consensus adhesome proteins can translocate to the nucleus and postulate that a subset of these – the LIM domain containing adhesions proteins can shuttle between the cell surface and the nucleus and may transduce mechano-sensory signals resulting in a modulation of gene expression. They also conclude that adhesome proteins may have key roles as nuclear proteins in cancer cells, although the manuscript shows no direct evidence of this other than the chosen cell line is a cancer cell line.

The manuscript is well written and describes a large body of work and could be a useful resource for cell biologists and describes interesting observations that require following-up further by the community.

I consider, however, that key experimental evidence is missing that underlie most of the experimental work presented in this manuscript. Without this evidence it is difficult for me to support the authors' findings. Some of the conclusions that they make are also extremely tenuous and hence in its current form this manuscript is a long way for being in a form suitable for publication.

Major concerns.

1. The whole basis of the study is that a 'purified' nuclear is extracted devoid of any non-nuclear contamination. The way in which the authors prove this to be the case is by westerns and quantitative proteomics followed by hierarchical clustering of data. From the westerns it is not clear why a Golgi protein is perinuclear and the ER marker protein is not. It would have been better to use markers of membrane proteins for these compartments. Indeed the cellular compartment data shown in supplementary figure 2 suggest a hefty dose of cellular membrane contamination in the nuclear fraction. Moreover, a plasma membrane integral membrane protein marker seems to be missing. Given that the fractionation is based on detergent solubilisation, showing no enrichment/contamination from other intracellular membrane is essential, especially as the adhesome is associated with the plasma membrane. The hierarchical clustering and quantitative proteomics data depiction is also questionable. The fractions taken for mass spec. quantitation and indeed immune-blotting were normalized by amount not proportion of the cell. Thus if the cytoplasmic fraction represented most of the total proteins, then a smaller proportion of it would have been taken for analysis. If, for example, small amounts of cytoplasmic contamination were present in the 'nuclear' fraction (no subcellular fractionation method ever gives a completely pure preparation –so this is likely) then the amount of the contamination would manifest as a strong signal, which would be marked in the case of abundant proteins. The necessary checks and balances need to be carried out here, with data normalized by total amount of protein in each fraction. Hierarchical clustering can cover a multitude of sins. I would have liked to have seen all the data displayed in terms of subcellular location – i.e. where are the membrane proteins from other subcellular regions and some better clustering tool used to show which sets of adhesome proteins that behave in similar ways to one another.

To my understanding the authors would like the readers to believe that a subset of adhesome proteins are also present in the nucleus, but the data as portrayed do not really show this. Figure 2d is just an indication with no substance and figure 2f is difficult to interpret in terms of mixed localization.

Together with normalization issues and the potential contamination of the nuclear fraction with other membranes, the data certainly do not lead the reader to believe that a 'pure' nuclear fraction is derived, and the mixed locations of adhesome proteins are not shown.

Without better evidence, a substantial proportion of the rest of the findings in this manuscript are open to question. The manuscript is woefully lacking in any microscopy data to back up the authors claims.

2. I assume that the cells were not synchronized. What effect would mitotic cells have on the overall distribution of adhesome proteins between nucleus and cytoplasm?

3. Supplementary figure 2f is supposed to show presence of adhesion proteins in the nucleus of other cell types. From what I can see, there are only two other cell types and the westerns are highly irreproducible between cell lines and are not replicated within each cell line – how is the reader supposed to believe this statement from the data presented?

4. What is the difference between stringently identified, high stringency identifications and other identified proteins?

5. The manuscript is devoid of microscopy images – why were novel nuclear localized proteins not validated by microscopy methods.

6. On page 12 the authors state that the majority of consensus adhesome proteins from four defined modules were identified in non-nuclear fractions whereas the protein unassigned to one of these modules in the literature but detected in adhesion complexes were enriched in the nuclear fraction. How confident are the authors that these latter proteins represent contamination in the existing datasets? They go onto suggest that they may be present in low abundance in adhesion complexes and have moonlighting function. This is an unsubstantiated claim as no attempts are made by microscopy or any other method to try and determine the proportion of these proteins in the nucleus compared with the cytoplasm.

7. FAK:

A. The authors go onto determine the focal adhesion kinase dependent nuclear proteome. Figure 3 e is confusing – why is there still FAK in the FAK^{-/-} cells. FAK-NLS is depleted (but according to figure 3e – not markedly) but the authors do not show its overall cellular abundance. If it is unstable and partially degraded, what they may be showing is a drop in the overall level levels of this protein results in its reduced contamination from the cytoplasm in the nuclear fraction. Associated microscopy images would dissuade me from my argument that apparently nuclear FAK is just a cytoplasmic contamination. The other two fractions (cytoplasmic and perinuclear) as well as the total cell lysate should also be blotted for FAK in the FAK^{-/-}, FAK-WT and FAK-NLS cells.

The authors give little insight into its mechanism of action in modulating expression of a sub-set of the nucleo-adhesome. They claim to have additional data regarding its formation with of a complex with IL33 in the nucleus. It would be useful to add this as at the moment, the results presented in this section are open to some speculation.

If FAK has another function in the nucleus to its function at focal adhesion why didn't the authors go some way to prove the need of an active kinase domain?

8. ILK: The authors then determine the interactions of another adhesome kinase, ILK by proximity tagging and immune precipitation although the two datasets are not compared for overlap. How do the authors know that the BirA tagged version is getting into the nucleus, and even if it is, how do they know that the majority of the biotinylation events are not happening by virtue of the cytoplasmic fraction?

Figure 4j is particularly worrying. The authors would like us to believe that there is less ILK in the nucleus when its putative NLS is mutated, but there is less ILK overall a fact which is overlooked by the authors. Unless the data are portrayed quantitatively, an alternative explanation could be that ILK is destabilised and degraded by upon mutation – hence I am not convinced – again without back up microscopy, that the authors have identified its NLS.

Minor concerns:

1. Although the authors focus on SCC cells they do not show that the nucleo-adhesome is any different in non-cancer cell lines, hence I think the title of the paper is mis-leading.
2. Figure 1d is unclear. It is not described in the main text. What does 'other' refer to in the key?
3. I do not understand the two major clusters in figure 2f – i.e. the left hand and right hand clusters? Why do some nuclear proteins have a positive Z core and other a negative Z-score? The significance of the Z-score and how it was calculated should be better described.
4. What do the two principle components correspond to in figure 3c?
5. Why did the authors only mutate some of the basic amino acids that could be involved in the NLS? It reads as if they chose 4 at random and got lucky (or not – see comments above).

Reviewer #2 (Remarks to the Author):

Please find below some comments related to the general objectives and strategy of the paper (NCOMMS-20-23531):

1. There is a strong emphasis, in the paper (and in the title) on the cancer relevance of the work. In my opinion, this aspect is not sufficiently addressed here. The fact that FAK is present in the nucleus of SCC cells, but not in WT keratinocytes, seems to me to be insufficient for justifying far-reaching conclusions about the cancer-specific nature of the nuclear localization of adhesome proteins.
2. The paper particularly focuses on FAK and ILK. The reason for choosing FAK is clear, given previous data on its role in immune evasion and other processes, but the choice of ILK is less obvious and not sufficiently addressed in the paper. Also - its role in regulating cell viability (with or without stress) is not very conclusive, in my opinion.
3. The 'functional rationale' is based on comparing the properties (e.g. viability) of KO SCC cells that are reconstituted with the WT protein (that localizes to the nucleus) or mutant protein (that fails to localize to the nucleus). One possibility that is not addressed, as far as I can tell, is that the mutation,

in addition to its effect on the entry to the nucleus, also changes the molecular interactions of the relevant proteins in the adhesion site. Such possibility should be discussed.

4. One last point (not a criticism) that I would like to add is an idea that could be considered (and may be relevant or even obvious to Margaret Frame), is that integrin adhesions are hot spots for tyrosine phosphorylation, driven by src, FAK and different RTKs, and the level of phosphorylation can be, and often is, affected by malignant transformation. Given indications that import to the nucleus, can be modulated by the state of phosphorylation, I wonder whether the level of tyrosine phosphorylation in the fraction of adhesome proteins residing in the nucleus ("The nuclear adhesions") was checked.

Reviewer #3 (Remarks to the Author):

In the submitted manuscript Byron et al., at continuing to explore the exciting world on nuclear adhesion proteins, demonstrated in detail for FAK by the Frame lab, but remaining otherwise poorly understood and under investigated. Here the authors have undertaken a massive effort to define the nuclear proteome of SCC cells, analyse it with respect to presence of adhesome proteins and investigate its FAK dependency. The authors find a striking 70% of the meta-adhesome protein in their nuclear fractions by mass-spec. They find that FAK regulates nuclear proteins mainly through impacting on their transcription rather than controlling their nuclear localization. The authors focus on ILK, an established key signaling and scaffolding hub in ECM adhesions. Interestingly, in contrast to FAK ILK did not markedly regulate expression or nuclear localization of other proteins. However, the authors suggest that nuclear ILK supports cell survival upon stress.

This is a massive set of carefully conducted biochemical+ proteomics+bioinformatics experimenta/data. These will be an important resource and inspiration for scientists working indifferent areas of cell biology. This warrant the somewhat limited functional data included in the manuscript. However, some more work is needed with the nuclear-ILK to validate the claims made by the authors.

Specific points.

Figure 2b. The fractionations look very convincing. However, it would be nice to see also how FAK partitions to different fractions with blotting (currently only the FAK localization to the nucleus fraction is shown in Figure 3 for the scc cells).

Figure 4j. The reduction in the nuclear localization of the ILK-NLS mutant is rather modest. Could the authors analyse this more convincingly using quantification from multiple experiments? The small difference compared to the wild-type and the almost identical interactomes based on using LC-MS/MS raises the question about the functionality of this mutation. Could it be that the reduced

survival of the ILK-NLS reconstituted cells is due to the lower expression levels evident in 4j? Are there other challenges than serum withdrawal where ILK confers increased survival? Anoikis?

Reviewers' comments:

Reviewer #1 (Remarks to the Author):

This manuscript focuses round the study of a subset of proteins involved in the formation of cell adhesion proteins that exist also within the nucleus in an attempt to understand potential moonlighting and other non-canonical roles of these proteins. The authors first determine the likely subset of nuclear localized adhesion proteins using informatics approaches on existing datasets, and create a list of adhesome proteins likely in associate with the nucleus.

The authors then use a detergent based subcellular fraction method to enrich for nuclear, perinuclear and cytoplasmic proteins in a squamous cell carcinoma (SCC) cell line and carry out quantitative mass spectrometry to determine enrichment of adhesome and associated proteins within each fraction and identify some of the above adhesome proteins as present in the nuclear fraction. Using a stringent list of these proteins they construct a network and describe sets of proteins with hitherto ascribed nuclear location from known adhesome proteins.

The authors then concentrate on two kinases, FAK and ILK, that are known to associate with the adhesome and attempt to determine the nuclear function of these proteins. In the case of FAK they assign a function in expression of a subset of cell surface or cytoskeletal proteins. Using a proximity tagging and immune-precipitation strategies, they determine interacting partners of ILK and find that a substantial proportion of interacting partners have the potential to locate within the nucleus. Knock-outs of ILK did not affect the nuclear proteome but the authors were able to demonstrate that cell survival was impaired in ILK knock-out cell lines.

In conclusion, the authors postulate that 70% of core consensus adhesome proteins can translocate to the nucleus and postulate that a subset of these – the LIM domain containing adhesions proteins can shuttle between the cell surface and the nucleus and may transduce mechano-sensory signals resulting in a modulation of gene expression. They also conclude that adhesome proteins may have key roles as nuclear proteins in cancer cells, although the manuscript shows no direct evidence of this other than the chosen cell line is a cancer cell line.

The manuscript is well written and describes a large body of work and could be a useful resource for cell biologists and describes interesting observations that require following-up further by the community.

I consider, however, that key experimental evidence is missing that underlie most of the experimental work presented in this manuscript. Without this evidence it is difficult for me to support the authors' findings. Some of the conclusions that they make are also extremely tenuous and hence in its current form this manuscript is a long way for being in a form suitable for publication.

Response

We thank the reviewer for their constructive comments and are pleased that they find our work interesting, well presented and a valuable resource for the research community. We have responded to all of the reviewer's concerns and have generated additional key experimental evidence to further support our findings, as detailed below:

Major concerns.

1. The whole basis of the study is that a ‘purified’ nuclear is extracted devoid of any non-nuclear contamination. The way in which the authors prove this to be the case is by westerns and quantitative proteomics followed by hierarchical clustering of data. From the westerns it is not clear why a Golgi protein is perinuclear and the ER marker protein is not. It would have been better to use markers of membrane proteins for these compartments.

Response

Our validation approach used a panel of subcellular markers, some with restricted intracellular localisation and others with more extensive reported localisations, to capture a range of expected marker enrichment in one or multiple subcellular fractions. Importantly, this included organellar membrane-associated proteins. **GM130** is a peripheral Golgi membrane protein, strongly attached to the Golgi membrane, mainly at the *cis*-Golgi side, proximal to the ER [Nakamura et al. (1995) *J. Cell Biol.* **131**, 1715–1726]; **RCAS1** is a type III transmembrane Golgi protein [Engelsberg et al. (2003) *J. Biol. Chem.* **278**, 22998–23007]. Both of these Golgi membrane proteins were thus annotated as “perinuclear region” subcellular markers (brown colour bars, **Fig. 2b**), and results using our subcellular fractionation protocol supported this classification. **ERp72**, in addition to residing in the ER, has been described to localise also at the cell surface [Weisbart et al. (1992) *J. Immunol.* **148**, 3958–3963], suggesting that it can be exported to the plasma membrane, which is why we annotated this protein with the “perinuclear region” (brown) and “cytoplasm/plasma membrane” (yellow) subcellular marker colour bars (**Fig. 2b**). This reported subcellular partitioning also fits with our observations of ERp72 in dual fractions by subcellular fractionation.

However, we appreciate that the reviewer wished for further analysis (see **Action taken** here and below), which we were happy to address.

Action taken

To expand the panel of membrane-associated protein markers for these compartments, we have now also examined the enrichment of the *trans*-Golgi network membrane protein **golgin-97**. Golgin-97 was consistently detected in the perinuclear/organellar fractions and was not detected in the nuclear fractions, providing important additional evidence that non-specific organellar membrane proteins do not contaminate our nuclear fractions. These data have been added to a new **Supplementary Fig. 2a**, which details more extensive characterisation of marker proteins in the subcellular fractions (additional new markers detailed below), and we have described these new data in the text (**Results, p. 6**).

Indeed the cellular compartment data shown in supplementary figure 2 suggest a hefty dose of cellular membrane contamination in the nuclear fraction. Moreover, a plasma membrane integral membrane protein marker seems to be missing. Given that the fractionation is based on detergent solubilisation, showing no enrichment/contamination from other intracellular membrane is essential, especially as the adhesome is associated with the plasma membrane. The hierarchical clustering and quantitative proteomics data depiction is also questionable.

Response

We further examined the over-representation analysis in Supplementary Fig. 2 and found that a substantial proportion of the proteins that contributed to apparent “contaminating” non-nuclear terms have actually also been reported to localise to the nucleus. For example, in an analysis of Gene Ontology (GO) term membership, we found that 56% of proteins in our nuclear fraction annotated as “cytosolic part” are also annotated as *nuclear* (AmiGO 2, GO term GO:0005634) and 31% of proteins annotated as “organelle inner membrane” are also annotated as *nuclear*, including the inner nuclear membrane proteins **emerin**, **Man1**, **SUN1** and **SUN2**. Thus, a proportion of the proteins annotated with “contaminating” terms are also known to localise to the expected subcellular compartment. This highlights a common challenge in quantifying and statistically testing the expected absence of individual cellular component annotations when multiple subcellular localisations may exist for many, or indeed every, protein. We do not believe this confounds our conclusions.

Action taken

We performed under-representation analysis, which suggested that the nuclear subproteome was depleted of plasma membrane and extracellular proteins [Reviewer Fig. 1 below (not included in the manuscript)]. Owing to the challenges in deconvoluting and interpreting *absence* of protein annotation terms bioinformatically (as detailed above), we have omitted the cellular component analysis panel of Supplementary Fig. 2 to avoid confusion.

GO:CC		stats		
Term name	Term ID	P _{adj}	$-\log_{10}(P_{adj})$	Show evidence codes
extracellular region	GO:0005576	9.342×10^{-31}		
extracellular space	GO:0005615	2.977×10^{-26}		
intrinsic component of plasma membrane	GO:0031226	5.835×10^{-25}		
integral component of plasma membrane	GO:0005887	1.603×10^{-23}		
external side of plasma membrane	GO:0009897	7.436×10^{-9}		
keratin filament	GO:0045095	7.965×10^{-5}		
intrinsic component of synaptic membrane	GO:0099240	2.348×10^{-4}		
integral component of synaptic membrane	GO:0099699	2.685×10^{-4}		
intrinsic component of postsynaptic membrane	GO:0098936	8.166×10^{-4}		
integral component of postsynaptic membrane	GO:0099055	2.240×10^{-3}		
intrinsic component of postsynaptic specialization mem...	GO:0098948	4.033×10^{-3}		
cation channel complex	GO:0034703	7.440×10^{-3}		
integral component of postsynaptic specialization mem...	GO:0099060	7.936×10^{-3}		
cell surface	GO:0009986	1.362×10^{-2}		
ion channel complex	GO:0034702	2.029×10^{-2}		
intrinsic component of postsynaptic density membrane	GO:0099146	2.446×10^{-2}		

1 to 16 of 16 <> Page 1 of 1 >>

Reviewer Figure 1 Cellular component under-representation analysis of the nuclear subproteome. Analysis performed using g:Profiler (version e101_eg48_p14_baf17f0). FDR-corrected P -values (p_{adj}) shown for all depleted terms (FDR < 0.05).

Instead, to verify the partitioning of plasma membrane-associated proteins (in addition to **ERp72** and **GAPDH**, which have been reported to localise at the cell surface and are not detectable in our nuclear fraction by immunoblotting, as shown in **Fig. 2b**), we probed subcellular fractions for hepatocyte growth factor receptor (**c-Met**), which is reported to localise to the plasma membrane and cytosol (The Human Protein Atlas, entry ENSG00000105976-MET). c-Met was detectable in the cytoplasmic and perinuclear/organelle fractions, but importantly **not** in the nuclear fraction. These data confirm the lack of a cell-surface-localised protein in the nuclear fraction and have been added to the new **Supplementary Fig. 2a**.

To examine the localisation of proteins associated with other pools of cellular membrane, we tested markers of different vesicle populations. We probed subcellular fractions for the very early endosomal membrane receptor adapter **APPL1**, the early endosomal membrane tethering factor **EEA1** and the lysosomal protease **cathepsin B** (which can also associate with the cell surface [Linke et al. (2002) *J. Cell Sci.* **115**, 4877–4889]). APPL1 and EEA1 are enriched on distinct populations of endosomes, and we observed that APPL1 was in the cytoplasmic fraction, whereas EEA1 was in the perinuclear/organellar fraction. Cathepsin B was detected in the cytoplasmic fraction. **Importantly, none of these membrane markers was detected in our nuclear fraction, providing further evidence indicating that the nuclear fraction is not contaminated with cellular membrane.** These data have been added to the new **Supplementary Fig. 2a**.

To complement our examination of the localisation of a marker of microtubules (**α -tubulin, Fig. 2b**), we also tested a marker of actin filaments, actin filament-associated protein 1 (**AFAP1**). AFAP1 was not detected in our nuclear fraction, indicating that this actin-associated protein, which can also localise at focal adhesions, does not partition to the nuclear fraction. This has been added to the new **Supplementary Fig. 2a** and described in the text (**Results, p. 6**).

Together, these data provide high confidence that non-specific membrane-associated (and cytoskeletal) components were not enriched in our nuclear fraction. We have expanded the relevant part of the text to clarify that non-nuclear membrane proteins do not contaminate the nuclear preparations (**Results, p. 6**).

The fractions taken for mass spec. quantitation and indeed immune-blotting were normalized by amount not proportion of the cell. Thus if the cytoplasmic fraction represented most of the total proteins, then a smaller proportion of it would have been taken for analysis. If, for example, small amounts of cytoplasmic contamination were present in the 'nuclear' fraction (no subcellular fractionation method ever gives a completely pure preparation –so this is likely) then the amount of the contamination would manifest as a strong signal, which would be marked in the case of abundant proteins. The necessary checks and balances need to be carried out here, with data normalized by total amount of protein in each fraction. Hierarchical clustering can cover a multitude of sins. I would have liked to have seen all the data displayed in terms of subcellular location – i.e. where are the membrane proteins from other subcellular regions and some better clustering tool used to show which sets of adhesion proteins that behave in similar ways to one another.

Response and action taken

Recent analysis of the subcellular map of the human proteome indicated that the nucleus (6,245 proteins) and the cytoplasm (4,279 proteins) have the largest, and relatively similarly sized, organellar subproteomes in the cell [Thul et al. (2017) *Science* **356**, eaal3321]. We had therefore normalised our samples by protein amount to enable accurate and comparable mass spectrometric analysis of the same amount of peptides in each sample. However, we agree that this does not necessarily reflect the proportional contribution to the cell of each organelle/fraction. We have thus re-normalised the subcellular proteomic profiling data as suggested by the reviewer to account for the original proportion of protein obtained from each fraction, and these re-analysed data have been incorporated into the new Figures,

Supplementary Figures and Supplementary Tables. In addition, our re-analysis of the data now required protein quantification in all biological replicates and with a >5% fraction of the cellular pool for the nuclear subproteome to ensure more stringent identification of proteins in the nuclear fraction and removal of potential nonspecific contamination.

To complement hierarchical cluster analysis of the subcellular proteomic profiling data, we have used t-distributed stochastic neighbour embedding (t-SNE) to visualise the dataset in low-dimensional feature space. We obtained a validated set of mouse subcellular marker proteins, including membrane proteins, using the R package pRoloc [Gatto et al. (2014) *Bioinformatics* **30**, 1322–1324] and curated these against the UniProt knowledgebase, Gene Ontology resource and the Human Protein Atlas to remove markers with conflicting or multiple reported subcellular locations. Mapping of these curated markers, as well as adhesome proteins, onto t-SNE plots enabled us to identify the partitioning of subcellular marker proteins that behave in similar ways in the dataset. These data are presented in a new **Fig. 2g** and **Supplementary Fig. 2d**. These analyses classify the data more fully by curated subcellular location, as requested by the reviewer, and show that a subset of adhesome proteins partition with nuclear proteins.

To my understanding the authors would like the readers to believe that a subset of adhesome proteins are also present in the nucleus, but the data as portrayed do not really show this. Figure 2d is just an indication with no substance and figure 2f is difficult to interpret in terms of mixed localization.

Response and action taken

The aim of Fig. 2d is to indicate the extent of detection of nuclear proteins in other subcellular fractions and, in addition, the detection in the nucleus of some cytoplasmic or perinuclear proteins. This agrees with a recent analysis, in which half of all proteins were found to be localised to multiple subcellular compartments, indicating that there is a shared pool of proteins even among functionally unrelated organelles [Thul et al. (2017) *Science* **356**, eaal3321]. We do not derive any conclusion about adhesome proteins from Fig. 2d but rather include it so as to: (i) summarise the numbers of proteins identified in each fraction by mass spectrometry and (ii) **support the concept that some proteins exist in multiple subcellular locations**, which we go on to investigate for adhesome proteins. For greater detail, we quantify this phenomenon more comprehensively in the complementary **Supplementary Fig. 2c**. We have added reference in the text to the study by Thul et al. (2017), which supports our rationale for analysing non-canonical localisation of adhesome proteins (**Results, p. 6**).

The mixed localisation of adhesome proteins in **Fig. 2f** shows many adhesome proteins, while having previously been described associated with integrin adhesome complexes, are also detected in the nucleus. This is further supported and resolved by the new t-SNE analysis (new **Fig. 2g** and **Supplementary Fig. 2d**). As detailed above, our proteomic analyses identified some proteins in multiple subcellular locations, which agrees with other studies [Thul et al. (2017) *Science* **356**, eaal3321] and is therefore not unexpected or controversial; what we show here for the first time is the **scale and nature of adhesome protein localisation in different subcellular fractions**.

Together with normalization issues and the potential contamination of the nuclear fraction with other membranes, the data certainly do not lead the reader to believe that a 'pure' nuclear fraction is derived, and the mixed locations of adhesion proteins are not shown. Without better evidence, a substantial proportion of the rest of the findings in this manuscript are open to question. The manuscript is woefully lacking in any microscopy data to back up the authors claims.

Response and action taken

As detailed above, we have re-analysed the subcellular proteomic profiling data using re-normalised mass spectrometry data with higher-stringency identification thresholds. We have also expanded our subcellular marker antibody panel to confirm lack of non-specific organellar membrane contamination. We agree with the reviewer's comment above that no subcellular fractionation method can ever be guaranteed to give a completely pure preparation – probably there is no such thing – but we have taken measures (detailed above) to ensure that we have derived nuclear fractions strongly enriched in nuclear proteins and depleted of potentially contaminating regions of the cell, including the perinuclear region, organellar membrane and cytoskeleton. This approach provides undeniable evidence of the scale and nature, based on biochemistry, of classical adhesion protein localisation to the nucleus; as such, we believe it is important new information and a useful resource for the community.

We have also performed microscopy experiments to confirm the nuclear localisation of several adhesion proteins, which we provide in new figures – **Fig. 2h**, **Fig. 3e** and **Fig. 5e** – and which we detail further in our response to major point 5 (below). This includes the demonstration by microscopy that the adhesion proteins paxillin, Hic-5 and testin localise to the nucleus. Testin has not been shown to localise to the nucleus before, so this is novel (whereas the other proteins have in different contexts, hence we had not included them in the first submission).

2. I assume that the cells were not synchronized. What effect would mitotic cells have on the overall distribution of adhesion proteins between nucleus and cytoplasm?

Response

The reviewer is correct, the cells were not synchronised, and we have not specifically addressed mitotic cells.

3. Supplementary figure 2f is supposed to show presence of adhesion proteins in the nucleus of other cell types. From what I can see, there are only two other cell types and the westerns are highly irreproducible between cell lines and are not replicated within each cell line – how is the reader supposed to believe this statement from the data presented?

Response and action taken

The data in Supplementary Fig. 2f suggested that adhesion proteins, albeit different sets of adhesion proteins, are present in cell types other than the cancer cells we used as a model here. We have removed these data since these cell types had not been analysed in anything like the same depth – we were just making the point that nuclear localisation of adhesion

proteins is not restricted to these cells. Since this is clear from the work of others, we have removed this panel of Supplementary Fig. 2.

4. What is the difference between stringently identified, high stringency identifications and other identified proteins?

Response and action taken

The first two terms mentioned by the reviewer described the same subset of proteins, so to improve clarity, we have revised our terminology to now only refer to “high-stringency identifications” (or proteins identified with high stringency) to describe those detected in all five biological replicates. This re-analysis provides a higher stringency [now requiring quantification in all biological replicates (previously, four out of five biological replicates) and with a >5% fraction of the cellular pool] to ensure more robust detection of the core nucleo-adhesome. Regardless of classification, all proteins were identified with FDR < 1% (**Methods, p. 22**). We have added a sentence to the text to clarify our classification of high-stringency identifications (**Results, p. 7**).

5. The manuscript is devoid of microscopy images – why were novel nuclear localized proteins not validated by microscopy methods.

Response and action taken

We had not shown microscopy images in our original submission because others have shown the presence of some individual adhesome proteins in the nucleus of other cells in a variety of different contexts. Therefore, we did not wish to be repeating others. However, in response to the reviewer, we now include some microscopy experiments to confirm the nuclear localisation of several adhesion proteins in our hands also. We verify, and quantify, by microscopy the nuclear localisation of paxillin and Hic-5, which, in particular, can be seen to accumulate in the nucleus upon inhibition of exportin-1 using leptomycin B (LMB), now provided in a new **Fig. 2h and i** and **Fig. 5e and f**. Moreover, we demonstrate the nuclear localisation of the adhesion protein testin, which we identified by nucleo-adhesome network analysis; this is novel, and we observed the accumulation of testin in the nucleus in response to H₂O₂-induced stress. These data are presented in a new **Fig. 3e and f**.

6. On page 12 the authors state that the majority of consensus adhesome proteins from four defined modules were identified in non-nuclear fractions whereas the protein unassigned to one of these modules in the literature but detected in adhesion complexes were enriched in the nuclear fraction. How confident are the authors that these latter proteins represent contamination in the existing datasets? They go onto suggest that they may be present in low abundance in adhesion complexes and have moonlighting function. This is an unsubstantiated claim as no attempts are made by microscopy or any other method to try and determine the proportion of these proteins in the nucleus compared with the cytoplasm.

Response and action taken

As mentioned above, like others, we believe that many proteins have functions in multiple subcellular locales; what we show here is the scale of the nuclear translocation of classical adhesion proteins, including those from three of the four modules of the consensus

adhesome, and a new protein testin. With the re-normalised MS data, we now provide protein intensity data scaled across the subcellular fractions to enable comparison of relative protein abundance (**Fig. 3b–d, Supplementary Fig. 3b and Supplementary Table 3**), and these data clearly show that the detected non-canonical nuclear adhesion proteins are relatively enriched in the nucleus compared to the other subcellular fractions.

In addition, using confocal microscopy, we observed that one of the non-canonical adhesome components we detected, hnRNP Q, is predominantly nuclear, with some cytoplasmic staining, in some cases in the vicinity of paxillin-positive membrane protrusions (**Reviewer Fig. 2** below). This is consistent with evidence from multiple cell lines presented in the Human Protein Atlas (entry ENSG00000135316-SYNCRIP), which shows detectable cytoplasmic and vesicular staining of hnRNP Q and, in some cases, localisation to membrane protrusions and ruffles (which are associated with cell migration and adhesion proteins) (**Reviewer Fig. 3** below). Indeed, Human Protein Atlas data show that several selected non-canonical adhesome components can localise to membrane protrusions and ruffles, in addition to more prominent localisation in the nucleus, suggesting that these proteins are not adhesome database contaminants and do localise to adhesion-associated subcellular structures (**Reviewer Fig. 3**).

To synthesise the evidence supporting this notion, we have plotted experimental and inferred evidence for subcellular localisation of each of the proteins not assigned an adhesome module. In addition, we have used deep learning-based sequence prediction (DeepLoc recurrent neural network algorithm) to assess the likelihood that each non-canonical adhesome protein can localise to various subcellular regions. These data are presented in a new **Supplementary Fig. 3d**. The data reveal reported evidence and/or prediction of localisation of the majority of these proteins to the cell membrane or cytoplasm as well as their better-characterised localisation and functions in the nucleus. Nonetheless, as the respective cytoplasmic and nuclear functions of these proteins are not the focus of the current paper, we have toned down our speculation in the text that the non-canonical nuclear adhesion proteins may have more prominent roles in the nucleus than at adhesion complexes (**Results, p. 9**).

Reviewer Figure 2 Confocal imaging of subcellular localisation of hnRNP Q in SCC cells. Nuclei were detected using NucBlue. Inverted lookup tables were applied; in merged images, colocalisation of paxillin (green) and hnRNP Q (magenta) is represented by black regions. Arrowheads indicate examples of hnRNP Q localisation to

cytoplasm or membrane protrusions. Two fields of cells from one experiment are shown, representative of three independent experiments. Scale bars, 20 μm .

Reviewer Figure 3 Confocal imaging of selected non-canonical adhesome proteins from the Human Protein Atlas. Cell lines are indicated. Inverted lookup tables were applied; colocalisation of the protein of interest (magenta) and tubulin (cyan) is represented by blue regions. Arrowheads indicate examples of protein-of-interest localisation to plasma membrane protrusions, lamellipodia or membrane ruffles. Images available from v20.proteinatlas.org [Thul et al. (2017) *Science* 356, eaal3321].

7. FAK:

A. The authors go on to determine the focal adhesion kinase dependent nuclear proteome. Figure 3 e is confusing – why is there still FAK in the FAK^{-/-} cells. FAK-NLS is depleted (but according to figure 3e – not markedly) but the authors do not show its overall cellular abundance. If it is unstable and partially degraded, what they may be showing is a drop in the overall level levels of this protein results in its reduced contamination from the cytoplasm in the nuclear fraction. Associated microscopy images would dissuade me from my argument that apparently nuclear FAK is just a cytoplasmic contamination. The other two fractions (cytoplasmic and perinuclear) as well as the total cell lysate should also be blotted for FAK in the FAK^{-/-}, FAK-WT and FAK-NLS cells.

Response and action taken

There is no FAK present in FAK^{-/-} cells as we have described in many publications. The MS/MS count for all four FAK^{-/-} biological replicates was 0, as was the corresponding label-free quantification (raw MS data available at ProteomeXchange; <http://proteomecentral.proteomexchange.org>). To enable statistical analysis, missing values (including those for FAK in FAK^{-/-} cells) were imputed with statistically modelled random numbers that simulate values below the detection limit based on the respective measured data distribution [Tyanova et al. (2016) *Nat. Methods* **13**, 731–740]. This gives rise to the very low numbers for absent proteins. This process is described in the Methods.

The FAK-NLS mutant is expressed at similar overall levels as FAK-WT – this has also been reported previously [Serrels et al. (2015) *Cell* **163**, 160–173], and we now include immunoblotting data to verify this in a new **Fig. 4b**, with no evidence of FAK degradation. The nuclear localisation of FAK is now well established (papers cited in the text), and previous studies have blotted for FAK and FAK-NLS across subcellular fractions [Serrels et al. (2015) *Cell* **163**, 160–173; Canel et al. (2017) *Cancer Res.* **77**, 5301–5312], so we did not wish to repeat this again here. Furthermore, nuclear FAK has been shown to be present in chromatin-containing subcellular fractions [Serrels et al. (2015) *Cell* **163**, 160–173]. In response to the reviewer, we now also include blotting of chromatin extracts (new **Fig. 4a**) to demonstrate that FAK-NLS is absent from the chromatin fraction, whereas FAK-WT (as previously shown) is present.

The authors give little insight into its mechanism of action in modulating expression of a sub-set of the nucleo-adhesome. They claim to have additional data regarding its formation with of a complex with IL33 in the nucleus. It would be useful to add this as at the moment, the results presented in this section are open to some speculation.

If FAK has another function in the nucleus to its function at focal adhesion why didn't the authors go some way to prove the need of an active kinase domain?

Response and action taken

We are sorry the reviewer missed the reference for this: the data relating to the mechanism of FAK's regulation of IL-33 has been published previously [Serrels et al. (2017) *Sci. Signal.* **10**, eaan8355], with extensive characterisation of the formation of a complex with IL-33 and regulation and biological consequences, so it would not be relevant to repeat that here. However, we now refer to this and the paper more clearly in the text (**Results, p. 12**).

With regard to the need for an active kinase domain, this, too, has been published previously [Serrels et al. (2015) *Cell* **163**, 160–173], in which the need for an active kinase domain in FAK's nuclear function was demonstrated. This function is to scaffold transcription regulatory complexes, including via IL-33, and we have reported a novel mechanism for that involving FAK-mediated regulation of chromatin accessibility in the nucleus [Griffith et al. (2021) *Sci. Rep.* **11**, 229]. The point of the current paper was not to work out these FAK mechanisms – which we have already reported – but instead to show the scale and identity of adhesion proteins that have nuclear roles. Sorry if this was unclear. We have reworded the text to ensure that this is clear now and have made sure that the appropriate references are prominent in the text.

8. ILK: The authors then determine the interactions of another adhesome kinase, ILK by proximity tagging and immune precipitation although the two datasets are not compared for overlap. How do the authors know that the BirA tagged version is getting into the nucleus, and even if it is, how do they know that the majority of the biotinylation events are not happening by virtue of the cytoplasmic fraction?

Figure 4j is particularly worrying. The authors would like us to believe that there is less ILK in the nucleus when its putative NLS is mutated, but there is less ILK overall a fact which is overlooked by the authors. Unless the data are portrayed quantitatively, an alternative explanation could be that ILK is destabilised and degraded by upon mutation – hence I am not convinced – again without back up microscopy, that the authors have identified its NLS.

Response

Owing to restrictions in place during the ongoing Covid-19 pandemic, including limited access to laboratories and supply chain issues, we were unable to undertake the significant experiments needed to address the role of ILK in much more detail. Also, upon re-analysis of our subcellular proteomic profiling data (major point 1, above) and the decision to increase stringency of nuclear protein identification (major point 4, above), ILK no longer passed the high detection threshold for core nucleo-adhesome membership (although it was still part of the expanded network), and it clearly does localise to the nucleus (as also described by others). We have therefore decided to remove the data on nuclear ILK, and we instead focussed on the novel question of whether there were subadhesion complexes in the nucleus with adhesion proteins working together. Hence, we have used multiple methods to now show that FAK resides within a network of nuclear adhesion proteins and we report, as an exemplar, a novel reciprocal association of FAK with one member of this network, Hic-5, which is known to function as a transcription coregulator. A Hic-5 nuclear network overlaps with the FAK network, and both nuclear adhesion proteins are able to regulate the gene expression of a common subset of nuclear proteins that are dependent on FAK's nuclear localisation. Together, these new data, presented in a new **Fig. 5, Supplementary Fig. 5 and Supplementary Tables 9 and 10**, identify a novel nuclear-localised adhesion protein network co-dependency, and describe a physical and likely functional interaction that contributes to adhesion-related gene expression.

Minor concerns:

1. Although the authors focus on SCC cells they do not show that the nucleo-adhesome is any different in non-cancer cell lines, hence I think the title of the paper is mis-leading.

Response and action taken

We agree with the reviewer and have changed the title of the paper to “Characterisation of a nucleo-adhesome”, and we have removed emphasis in the text about the phenomenon being general to cancer cells.

2. Figure 1d is unclear. It is not described in the main text. What does ‘other’ refer to in the key?

Response and action taken

Fig. 1d shows functional clustering analysis of biological processes enriched in the adhesome. We had (in the previous manuscript) referred to this alongside Fig. 1c, which

shows related clustering of subcellular locations, but we now refer to these networks separately in the text for clarity (**Results, p. 5**). “Other” refers to additional functional subcategories within the main category of each of the three networks (cell adhesion, intracellular transport and nuclear functions), but we have now made this much clearer in **Fig. 1d** by annotating the key more fully.

3. I do not understand the two major clusters in figure 2f – i.e. the left hand and right hand clusters? Why do some nuclear proteins have a positive Z core and other a negative Z-score? The significance of the Z-score and how it was calculated should be better described.

Response and action taken

A new **Fig. 2f** has been generated with the re-normalised MS data. The new heatmap, clustered by Spearman rank correlation, is partitioned into main clusters based on a Spearman rank correlation coefficient threshold of 0.5. This data-driven partitioning enabled the separation of proteins relatively enriched in different subcellular fractions. We have replaced z-score here with protein intensity scaled across the subcellular fractions (0, minimum intensity; 1, maximum intensity), which is more intuitive to interpret for relative comparisons between subcellular locations.

4. What do the two principle components correspond to in figure 3c?

Response

The principal components displayed are the two orthogonal low-dimensional projections of the MS data that account for the greatest proportion of the variance in the dataset. This dimensionality reduction approach enables the assessment of the underlying structure and relative protein profiles within the high-dimensionality quantitative proteomic dataset. Although, as we mention in the text, there is limited separation in PCA space owing to only a small proportion of the nuclear subproteomes being differentially regulated between the different cell lines, principal component 1 in this figure (now **Fig. 4d**) distinguishes the samples by cell line, indicating that the greatest variance in the dataset is explained by the different status of FAK (FAK-WT, FAK^{-/-}, FAK-NLS).

5. Why did the authors only mutate some of the basic amino acids that could be involved in the NLS? It reads as if they chose 4 at random and got lucky (or not – see comments above).

Response

We have removed the data on nuclear ILK to enable us to focus instead on the more novel information on nuclear subadhesion networks and a new complex between FAK and another nuclear adhesion protein, Hic-5 (detailed above).

Reviewer #2 (Remarks to the Author):

Please find below some comments related to the general objectives and strategy of the paper (NCOMMS-20-23531):

1. There is a strong emphasis, in the paper (and in the title) on the cancer relevance of the work. In my opinion, this aspect is not sufficiently addressed here. The Fact that FAK is present in the nucleus of SCC cells, but not in WT keratinocytes, seems to me to be insufficient for justifying far-reaching conclusions about the cancer-specific nature of the nuclear localization of adhesion proteins.

Response and action taken

As mentioned above (answer to Reviewer #1, minor point 1), we have removed implication in the text about any general cancer-specific nature of the findings, and we have changed the title to “Characterisation of a nucleo-adhesome”.

2. The paper particularly focuses on FAK and ILK. The reason for choosing FAK is clear, given previous data on its role in immune evasion and other processes, but the choice of ILK is less obvious and not sufficiently addressed in the paper. Also - its role in regulating cell viability (with or without stress) is not very conclusive, in my opinion.

Response

We had chosen to end our paper with studies on ILK as an additional nuclear adhesion protein that lies at the heart of another of the modules of the consensus adhesome. However, for the reasons mentioned above (answer to Reviewer #1, major point 8), we have removed the data on nuclear ILK and we instead focussed on the novel question of whether there were subadhesion complexes in the nucleus with adhesion proteins working together. Hence, we have used multiple methods to now show that FAK resides within a network of nuclear adhesion proteins and we report, as an exemplar, a novel reciprocal association of FAK with one member of this network, Hic-5, which is known to function as a transcription coregulator. A Hic-5 nuclear network overlaps with the FAK network, and both nuclear adhesion proteins are able to regulate the gene expression of a common subset of nuclear proteins that are dependent on FAK’s nuclear localisation. Together, these new data, presented in a new **Fig. 5**, **Supplementary Fig. 5** and **Supplementary Tables 9 and 10**, identify a novel nuclear-localised adhesion protein network co-dependency, and describe a physical and likely functional interaction that contributes to adhesion-related gene expression.

3. The ‘functional rationale’ is based on comparing the properties (e.g. viability) of KO SCC cells that are reconstituted with the WT protein (that localizes to the nucleus) or mutant protein (that fails to localize to the nucleus). One possibility that is not addressed, as far as I can tell, is that the mutation, in addition to its effect on the entry to the nucleus, also changes the molecular interactions of the relevant proteins in the adhesion site. Such possibility should be discussed.

Response

See above response to point 2.

4. One last point (not a criticism) that I would like to add is an idea that could be considered (and may be relevant or even obvious to Margaret Frame), is that integrin adhesions are hot spots for tyrosine phosphorylation, driven by src, FAK and different RTKs, and the level of phosphorylation can be, and often is, affected by malignant transformation. Given

indications that import to the nucleus, can be modulated by the state of phosphorylation, I wonder whether the level of tyrosine phosphorylation in the fraction of adhesome proteins residing in the nucleus (“The nuclear adhesome”) was checked.

Response

We have not checked this – but it is a very interesting idea and we will address this beyond this paper. Very specific and potent drugs would enable this.

Reviewer #3 (Remarks to the Author):

In the submitted manuscript Byron et al., at continuing to explore the exciting world on nuclear adhesion proteins, demonstrated in detail for FAK by the Frame lab, but remaining otherwise poorly understood and under investigated. Here the authors have undertaken a massive effort to define the nuclear proteome of SCC cells, analyse it with respect to presence of adhesome proteins and investigate its FAK dependency. The authors find a striking 70% of the meta-adhesome protein in their nuclear fractions by mass-spec. They find that FAK regulates nuclear proteins mainly through impacting on their transcription rather than controlling their nuclear localization. The authors focus on ILK, an established key signaling and scaffolding hub in ECM adhesions. Interestingly, in contrast to FAK ILK did not markedly regulate expression or nuclear localization of other proteins. However, the authors suggest that nuclear ILK supports cell survival upon stress.

This is a massive set of carefully conducted biochemical+ proteomics+bioinformatics experiments/data. These will be an important resource and inspiration for scientists working in different areas of cell biology. This warrants the somewhat limited functional data included in the manuscript. However, some more work is needed with the nuclear-ILK to validate the claims made by the authors.

Response

We thank the reviewer for his/her comments and enthusiasm for our work. We have re-normalised our subcellular fractionation profiling MS data and increased the stringency of the cut-off (as mentioned above; answer to Reviewer #1, major points 1 and 4), which now results in the detection of 57% of the consensus adhesome in the nucleus of the cells we use here.

We have considerably expanded our functional follow-up data (see below). We now show that FAK resides within a network of nuclear adhesion proteins and discover a novel reciprocal association with Hic-5, which is known to function as a transcription coregulator. These new data identify a functional association and co-dependency within a novel nuclear-localised adhesion protein subcomplex that controls adhesion-related gene expression. We think this is an exciting and novel finding to finish the paper with.

Specific points.

Figure 2b. The fractionations look very convincing. However, it would be nice to see also how FAK partitions to different fractions with blotting (currently only the FAK localization to the nucleus fraction is shown in Figure 3 for the scc cells).

Response and action taken

We now include in supplementary data clear evidence of FAK in multiple subcellular fractions (**Supplementary Fig. 4a**).

Figure 4j. The reduction in the nuclear localization of the ILK-NLS mutant is rather modest. Could the authors analyse this more convincingly using quantification from multiple experiments? The small difference compared to the wild-type and the almost identical interactomes based on using LC-MS/MS raises the question about the functionality of this mutation. Could it be that the reduced survival of the ILK-NLS reconstituted cells is due to the lower expression levels evident in 4j? Are there other challenges than serum withdrawal where ILK confers increased survival? Anoikis?

Response and action taken

See responses to Reviewer #1 and Reviewer #2 above. Briefly, we had chosen to end our paper with studies on ILK as an additional nuclear adhesion protein that lies at the heart of another of the modules of the consensus adhesome. However, for the reasons mentioned above (answer to Reviewer #1, major point 8), we have removed the data on nuclear ILK and we instead focussed on the novel question of whether there were subadhesion complexes in the nucleus with adhesion proteins working together. Hence, we have used multiple methods to now show that FAK resides within a network of nuclear adhesion proteins and we report, as an exemplar, a novel reciprocal association of FAK with one member of this network, Hic-5, which is known to function as a transcription coregulator. A Hic-5 nuclear network overlaps with the FAK network, and both nuclear adhesion proteins are able to regulate the gene expression of a common subset of nuclear proteins that are dependent on FAK's nuclear localisation. Together, these new data, presented in a new **Fig. 5, Supplementary Fig. 5 and Supplementary Tables 9 and 10**, identify a novel nuclear-localised adhesion protein network co-dependency, and describe a physical and likely functional interaction that contributes to adhesion-related gene expression. As mentioned above, we think this is an exciting and novel finding to finish the paper with.

REVIEWER COMMENTS

Reviewer #1 (Remarks to the Author):

The authors in general have done a good job in responding to the majority of my concerns with the first version of the manuscript. The new version is much improved and contains new data that is much more supportive of their claims.

I understand completely the authors' decision to drop the nuclear ILK data due to limited amount of experimental work they have been able to achieve during the pandemic. I do not believe that this omission in any way weakens the revised manuscript.

Some issues with the manuscript still remain, however, that should be addressed before this work is fit for publication. My main concern is still the interpretation of the data presented in terms of the nuclear localization of key adhesion proteins.

In the revised version of the manuscript, the authors present data to convince the readers of their nuclear enrichment and localization of these proteins in a number of ways, some of which seem to be contradictory.

The westerns presented in figure 2b and supplementary figure 2a show no signal from a variety of proteins from other subcellular location in the nuclear fraction, but the LC-MS/MS data clearly showed signal for these proteins as far as I can tell from supplementary table 2 in the nuclear fraction. The MS data is likely to be much more informative than the westerns whose exposure levels could belie signal, therefore in Line 122 – please remove the word 'purified' – I suspect that the nuclei have not been purified nuclei, but enriched to a greater or less extent which is not altogether obvious from the data.

The underrepresentation data the authors report in their rebuttal shows 'depletion' not 'complete removal' of non-nuclear compartments

The methods section does not make clear what data have been used to create figure 2g, but I assume it is all the LC-MS data, covering all three enriched samples. The consensus adhesion circles are difficult to see, but having zoomed it appears that only a few cluster with the nucleus. The rest are scattered throughout the plot. When looking at suppl fig 2d, which also contains data from the mitochondria, it is clear that the method chosen for subcellular fractionation has done a poor job at resolving the nucleus from other compartments. I do not like data presented where a subset has been taken away (mitochondria in this case) to make the figure look nicer. This seems to be cherry picking. As many mitochondrial proteins seem to overlay with the nuclear cluster in suppl 2d, how can the authors know that the adhesion proteins in this area of the plot are not from mitochondrially associated? Moreover 'other organelles/membrane' are not portrayed clearly on figure 2g – for example, where are the PM markers?. Please replace Figure 2g with suppl. Figure 2d, as this is more honest depiction of the data and add markers from missing subcellular niches such as the plasma membrane.

The authors state 'curated marker proteins generally partitioned according to subcellular location' – this is qualitative statement based on visualisation alone, and in many cases I simply don't agree.

At best the spots three open circles that seem to cluster best with the nuclear cluster – I suggest that the names of these proteins are added to the figure. The rest are either in mixed location or better cluster with the cytosol or ER/Golgi. I am confused to where the meta-adhesome proteins that the authors confidently claim to be nuclear, are within the data shown in figure 2g. Why wasn't the curated dataset plotted rather than just the consensus? It very unclear where the curated adhesome highlighted in the nuclear cluster in figure 2f are located in 2g, or where the consensus adhesome proteins in the same cluster in 2f are located.

Additional plotting of these data should be considered.

What the authors should have considered is to apply some form of classification tools, using sets of markers for the major organelles - for example SVM analysis e.g. Itzhak, DN et al Global, quantitative and dynamic mapping of protein subcellular localization. *Elife* 5, (2016) or Geladaki A et al (2019) Combining LOPIT with differential ultracentrifugation for high-resolution spatial proteomics *Nature Comms* 10(1):331

In summary, from the data presented in figure 2 and suppl figure 2, it appears that some of the adhesome proteins may have a nuclear localisation, some are in mixed localisation but it is not clear how much of a nuclear localization contributes to 'mixed localization', and many have a steady state location in other parts of the cell.

The authors again claim

'Strikingly, we found that 71.3% (1,719 proteins) of the experimentally derived meta-adhesome was detected in nuclear fractions, with 1,212 meta-adhesome proteins (50.2%) stringently identified in all five biological replicate experiments '

But my concerns are still that the nuclear fraction is far from pure – the LC-MS/MS data shows us this and without some classification tools, the authors cannot say that these proteins are present in the nuclear fraction by virtue of contamination from other compartments.

They authors may have attempted to impose a thresholding method to filter those proteins with significant signal in non-nuclear fractions by

'For further analysis and thresholding of the dataset, we classified proteins identified in all five experiments, and quantified with a >5% fraction of the cellular pool, as high-stringency identifications. Meta-adhesome proteins identified in nuclear fractions with high stringency were enriched for proteins with nuclear functions and known nuclear localisation as compared to the total set of meta-adhesome components'

This is very unclear; the authors should have employed robust classification tools. Or at the very least described in more detail how this thresholding was performed and why a 5% fraction of the cellular pool was an appropriate threshold.

How were the values in Suppl table 3 calculated? Why has paxillin (the gold standard) got very little intensity in the nuclear fraction? This is also true for all the meta-adhesome proteins. The authors should be much clearer about how this scaling was carried out and what the reader should interpret from the data

To convince the reader further the authors go on to perform microscopy, but only on a few proteins, where are these proteins on figure 2g?

In summary, I still think the authors have a little work to do on the way in which their data are portrayed to convince the reader that the nuclear meta-adhesome proteins are nuclear and not present in their nuclear enriched fractions by virtue of contamination from other subcellular compartments.

Minor corrections:

1. The figure legend 2g points to another t-SNE plot as the supplementary fig. 2e. This should read 2d
2. It is not clear what the units are in supplementary table 2

REVIEWER COMMENTS

Reviewer #1 (Remarks to the Author):

The authors in general have done a good job in responding to the majority of my concerns with the first version of the manuscript. The new version is much improved and contains new data that is much more supportive of their claims.

I understand completely the authors' decision to drop the nuclear ILK data due to limited amount of experimental work they have been able to achieve during the pandemic. I do not believe that this omission in any way weakens the revised manuscript.

Some issues with the manuscript still remain, however, that should be addressed before this work is fit for publication. My main concern is still the interpretation of the data presented in terms of the nuclear localization of key adhesion proteins.

In the revised version of the manuscript, the authors present data to convince the readers of their nuclear enrichment and localization of these proteins in a number of ways, some of which seem to be contradictory.

The westerns presented in figure 2b and supplementary figure 2a show no signal from a variety of proteins from other subcellular location in the nuclear fraction, but the LC-MS/MS data clearly showed signal for these proteins as far as I can tell from supplementary table 2 in the nuclear fraction. The MS data is likely to be much more informative than the westerns who exposure levels could belie signal, therefore in Line 122 – please remove the work 'purified' – I suspect that the nuclei have not been purified nuclei, but enriched to a greater or less extent which is not altogether obvious from the data.

The underrepresentation data the authors report in their rebuttal shows 'depletion' not ;'complete removal' of non-nuclear compartments

Response and action taken

We are pleased that the reviewer found the revised manuscript much improved and more supportive of our claims.

We are happy with the reviewer's suggestion to change the terminology in the manuscript, which we have now incorporated in the text. We note that the LC-MS data presented in Supplementary Data 2 (formerly Supplementary Table 2), as referred to by the reviewer, consists of processed, normalised data for which missing values (where corresponding peptides were not detected) have been imputed to enable statistical analysis. Therefore, quantitative values for proteins that were absent from a given sample, or below the limit of quantification, and have no signal by MS, are imputed with values drawn from the empirical distribution of the data and transformed to simulate expression below the detection limit; thus, these present as very low intensities in Supplementary Data 2. The raw MS data, and an associated TXT file detailing all protein identifications, are available via the MS data repository ProteomeXchange (<http://proteomecentral.proteomexchange.org>), which we now link to in the manuscript, to enable further interrogation of proteins that were not detected in given subcellular fractions.

The methods section does not clear what data have been used to create figure 2g, but I assume it is all the LC-MS data, covering all three enriched samples. The consensus

adhesome circles are difficult to see, but having zoomed it appears that only a few cluster with the nucleus. The rest are scattered throughout the plot. When looking at suppl fig 2d, which also contains data from the mitochondria, it is clear that the method chosen for subcellular fractionation has done a poor job at resolving the nucleus from other compartments. I do not like data presented where a subset has been taken away (mitochondria in this case) to make the figure look nicer. This seems to be cherry picking. As many mitochondrial proteins seem to overlay with the nuclear cluster in suppl 2d, how can the authors know that the adhesome proteins in this area of the plot are not from mitochondrially associated? Moreover 'other organelles/membrane' are not portrayed clearly on figure 2g – for example, where are the PM markers?. Please replace Figure 2g with suppl. Figure 2d, as this is more honest depiction of the data and add markers from missing subcellular niches such as the plasma membrane.

Response and action taken

The reviewer is correct that the t-SNE analysis was performed on the entire LC-MS dataset, including all three enriched fractions (provided in Supplementary Data 2). We apologise for omitting this information from the Methods section, and we have now detailed this accordingly in the text (**Methods, p. 25**) and in the figure legend for Fig. 2. We have enlarged the adhesome circles in the t-SNE plot to improve the visualisation, and we have now included all the detailed subcellular classification in **Fig. 2g**, as requested by the reviewer.

With regard to the location of adhesome proteins in several areas of feature space in the t-SNE plot, we are cautious not to over-interpret t-SNE plots since the two-dimensional non-linear embedding is unlikely to capture all structure present in the underlying high-dimensional dataset and distances between clusters are not preserved by t-SNE. Nevertheless, the distribution of adhesome proteins noted by the reviewer likely reflects the putative moonlighting nature of their nuclear functions: rather than serving as the primary subcellular location for these proteins, a proportion of the total cellular adhesion protein pool dynamically shuttles to, or resides in, the nucleus to function there. In contrast, most of the curated nuclear markers defined in the subcellular classification (e.g. chromatin remodelling complex components, nucleolar proteins) are primarily, or almost exclusively, resident in the nucleus in a more stable manner. Indeed, the nuclear markers were specifically curated for their reported non-cytoplasmic localisation to avoid fuzzy nuclear protein classification. Therefore, the cluster of nuclear markers resolved by t-SNE is unlikely to encompass adhesion proteins that have nuclear and cytoplasmic pools. The nucleocytoplasmic shuttling transcriptional co-activator YAP1, cyclin-dependent kinase inhibitor p27 and transcription factor p65, for example, do not co-locate with the cluster of curated stable nuclear markers by t-SNE – although they do co-locate with a subset of adhesome proteins [**Reviewer Fig. 1** below (not included in the manuscript)]. Instead, we use this dimensionality reduction approach as an additional method to visualise the high-dimensional dataset and to verify that primary nuclear markers partitioned from markers of other subcellular locations, confirming our isolation of nucleus-enriched fractions.

Reviewer Figure 1 Distribution of nucleocytoplasmic shuttling proteins away from curated nucleus-resident markers. **(a, b)** t-SNE maps of the subcellular proteomes annotated with curated subcellular markers, selected nucleocytoplasmic shuttling proteins (labelled) and consensus adhesome proteins **(a)** or literature-curated adhesome proteins **(b)**.

As noted by the reviewer, a subset of mitochondrial markers were relatively co-located in feature space with nuclear markers in the t-SNE plot, reflecting published reports that some mitochondrial proteins reside also in the nucleus (reviewed in DOI: 10.1016/j.tibs.2015.10.003) or can be degraded there (DOI: 10.7554/eLife.61230), but it may also suggest contamination with a subset of mitochondrial proteins. We have added text and a reference (**Results, p. 7**) describing this potential overlap. As above, however, distances between neighbourhoods (e.g. clusters of marker proteins) are not always preserved by t-SNE, which may introduce some overlap of distant clusters. There is no evidence to our knowledge of mitochondrion-associated adhesion proteins, whereas several adhesion proteins have been reported to localise to, and function in, the nucleus (DOIs: 10.1016/j.ceb.2006.08.006, 10.1016/j.ceb.2016.02.013), prompting us to explore this concept at a proteome scale in this manuscript. We hope the reviewer will agree that we have corroborated this now for several nuclear adhesion proteins by imaging (Figs 2h,i, 3e,f, 5e,f) and functional experiments (Figs 4, 5, Supplementary Fig. 4).

The authors state ‘curated marker proteins generally partitioned according to subcellular location’ – this is qualitative statement based on visualisation alone, and in many cases I simply don’t agree.

At best the spots three open circles that seem to cluster best with the nuclear cluster – I suggest that the names of these proteins are added to the figure. The rest are either in mixed location or better cluster with the cytosol or ER/Golgi.

I am confused as to where the meta-adhesome proteins that the authors confidently claim to be nuclear, are within the data shown in figure 2g. Why wasn’t the curated dataset plotted rather than just the consensus? It very unclear where the curated adhesome highlighted in the nuclear cluster in figure 2f are located in 2g, or where the consensus adhesome proteins in the same cluster in 2f are located.

Additional plotting of these data should be considered.

Response and action taken

The reviewer is correct that the statement about marker proteins is qualitative, as is typical of assessments of non-linear t-SNE embedding. However, we were not making a case that curated adhesome proteins partitioned to specific subcellular locations (such as the nuclear

cluster), as the reviewer suggests, but instead we were making the case that most of the subcellular classes in Fig. 2g (e.g. cytosol, ribosome, proteasome, nucleus) were located in distinct regions of the t-SNE plot. We apologise for any confusion; to avoid ambiguity, we have removed the word “curated” from the statement (**Results, p. 7**) and replaced it with “subcellular” to clarify reference to subcellular marker proteins.

As discussed above, the distribution of adhesome proteins in the t-SNE plot likely reflects the multiple subcellular locales to which they traffic and/or localise dynamically, so we would not expect nuclear adhesion proteins to co-cluster necessarily with stably resident nuclear protein markers. To add further context of adhesion protein distribution, we have generated a t-SNE plot annotated with the identified literature-curated adhesome proteins, as suggested by the reviewer, which we include as a new **Supplementary Fig. 2d**. As also suggested by the reviewer, we have performed additional plotting to highlight the consensus adhesome proteins and literature-curated adhesome proteins that partitioned in the nuclear subcluster by hierarchical clustering (Fig. 2f), and we have added these data as a new **Supplementary Fig. 2e** (which is annotated with rigorous machine learning-based subcellular classification; see below). We have added description of these data to the text (**Results, p. 7**). Since t-SNE does not inherently generate cluster assignments, we have also mapped proteins partitioned in the nuclear subcluster by hierarchical clustering (Fig. 2f) onto **Supplementary Data 2** to enable cross-referencing of the data.

What the authors should have considered is to apply some form of classification tools, using sets of markers for the major organelles - for example SVM analysis e.g. Itzhak, DN et al Global, quantitative and dynamic mapping of protein subcellular localization. *Elife* 5, (2016) or Geladaki A et al (2019) Combining LOPIT with differential ultracentrifugation for high-resolution spatial proteomics *Nature Comms* 10(1):331

In summary, from the data presented in figure 2 and suppl figure 2, it appears that some of the adhesome proteins may have a nuclear localisation, some are in mixed localisation but it is not clear how much of a nuclear localization contributes to ‘mixed localization’, and many have a steady state location in other parts of the cell.

The authors again claim

‘Strikingly, we found that 71.3% (1,719 proteins) of the experimentally derived meta-adhesome was detected in nuclear fractions, with 1,212 meta-adhesome proteins (50.2%) stringently identified in all five biological replicate experiments ‘

But my concerns are still that the nuclear fraction is far from pure – the LC-MS/MS data shows us this and without some classification tools, the authors cannot say that these proteins are present in the nuclear fraction by virtue of contamination from other compartments.

The authors may have attempted to impose a thresholding method to filter those proteins with significant signal in non-nuclear fractions by

‘For further analysis and thresholding of the dataset, we classified proteins identified in all five experiments, and quantified with a >5% fraction of the cellular pool, as high-stringency identifications. Meta-adhesome proteins identified in nuclear fractions with high stringency were enriched for proteins with nuclear functions and known nuclear localisation as compared to the total set of meta-adhesome components’

This is very unclear; the authors should have employed robust classification tools. Or at the very least described in more detail how this thresholding was performed and why a 5% fraction of the cellular pool was an appropriate threshold.

Response and action taken

We have performed supervised machine learning to classify protein subcellular localisation using a support vector machine (SVM) algorithm, as suggested by the reviewer. To generate labelled training data, we incorporated conservative curation of subcellular marker proteins to identify a set of core organelle marker proteins that are resident in the nucleus, the cytoplasm or elsewhere (perinuclear/organellear/other) to reflect the resolution of the subcellular fractionation procedure and to remove marker classes with insufficient marker proteins for supervised learning. Details of the marker proteins used have been added to a new **Supplementary Data 11**. We assessed and optimised the free parameters of the classifier using a grid search, maximising classifier accuracy before prediction of protein class labels was performed (**Reviewer Fig. 2** below).

Reviewer Figure 2 Optimisation of SVM algorithmic performance. Labelled data were partitioned into training and testing subsets (80:20, training:testing) and performance was estimated with five-fold cross-validation, repeated 100 times, and summarised by the harmonic mean of precision and recall (F1 score). **(a)** Box plots summarising distributions of the 100 F1 scores for the best cost–sigma parameter pairs. The distribution of F1 scores for the cost 8 and sigma 1 parameter pair was below the minimum value of the y-axis. **(b)** Heat map of the average F1 scores for the full range of parameter pairs tested. For **a** and **b**, cost represents the cost of constraints violation and sigma represents the inverse kernel width for the radial basis kernel.

To resultant subcellular prediction scores derived from localisation classification, we applied a false-discovery rate threshold of 10% based on concordance with evidence for subcellular localisation in Gene Ontology, UniProt or Cell Atlas as well as the literature. The results of this classification indicate the most likely subcellular location of each protein. Importantly, while 67.7% of classified meta-adhesome proteins were predicted as cytoplasmic, 19.9% were predicted as nuclear. Although a proportion of the adhesome (and the detected proteome) was not predicted to locate to a single subcellular locale at the given statistical threshold, in agreement with a report that much of the proteome localises to multiple subcellular compartments (DOI: 10.1126/science.aal3321), these classification results identify a robust pool of nucleus-associated adhesome proteins with strong predicted nuclear localisation, further supporting our evidence of a subset of nuclear adhesion

proteins. We have plotted the results of the classification in a new **Supplementary Fig. 2e**, including highlighting adhesome proteins partitioned in the nuclear subcluster by hierarchical clustering (see above). We have also added details of this approach to the text (**Results, p. 7; Methods, pp. 26–27**) and incorporated the classification data into **Supplementary Data 2**.

How were the values in Suppl table 3 calculated? Why has paxillin (the gold standard) got very little intensity in the nuclear fraction? This is also true for all the meta-adhesome proteins. The authors should be much clearer about how this scaling was carried out and what the reader should interpret from the data

Response and action taken

The values in Supplementary Data 3 (formerly Supplementary Table 3) are protein intensities scaled across the subcellular fractions using an approach known as min-max scaling, where 0 is the minimum intensity across all samples and 1 is the maximum intensity across all the samples per protein (row-wise). For each protein (row), all values are thus transformed into the range [0,1]. Therefore, low scaled intensities represent samples in which the protein was quantified as relatively less abundant than in other samples, but not necessarily of low absolute abundance overall.

In the case of paxillin, its abundance in the nuclear fraction is lower than its abundance in the other subcellular fractions, suggesting that, at steady state, there is a low proportion of total cellular paxillin in the nucleus, which we nonetheless detect by LC-MS and which we corroborate using confocal microscopy in Fig. 2h. This agrees with a report that the primary location of paxillin at steady state is extranuclear (DOI: 10.1074/jbc.M109446200). The nucleocytoplasmic shuttling of paxillin can, however, be revealed by blocking its nuclear export, which leads to accumulation of paxillin in the nucleus (DOI: 10.1074/jbc.M109446200), and we observe a significant increase in nuclear paxillin with leptomycin B (exportin-1 inhibitor) treatment of SCC cells using confocal microscopy (Fig. 2h,i). Furthermore, the relative distribution of paxillin across subcellular fractions quantified by LC-MS is very similar to that of YAP1, a transcriptional co-activator regarded as an archetypical protein that shuttles between the cytoplasm and the nucleus (**Reviewer Fig. 3a,b** below). Like paxillin, YAP1 is of lower abundance in the nucleus than the other subcellular fractions, which may reflect minor nuclear pools of these proteins compared to other organellar pools or the averaging of transient accumulation of these nucleocytoplasmic proteins in the nucleus when measured across the population of fractionated cells.

Reviewer Figure 3 Quantification of relative abundance of proteins identified by LC-MS in subcellular fractions. (a–d) Quantification of paxillin (a), YAP1 (b), α -actinin-1 (c) and ponsin (d) by label-free quantification (LFQ). Black bars, mean; light grey box, range. $n = 5$ independent biological replicates.

It is not the case, however, that all meta-adhesome proteins have low relative abundance in the nucleus, as commented upon by the reviewer. For example, the consensus adhesome proteins α -actinin-1 and ponsin are of higher abundance in the nucleus than the other subcellular fractions (**Reviewer Fig. 3c,d**). Indeed, as shown in Fig. 2f and detailed in Supplementary Data 3, many adhesome proteins in our dataset have high relative abundance in the nucleus compared to other subcellular fractions. To illustrate this point further, we plotted the distributions of scaled intensities (i.e. abundance relative to the other subcellular fractions) for all meta-adhesome proteins (**Reviewer Fig. 4** below). These data indicate that, while some adhesome proteins are detected at relatively low levels in the nucleus, a substantial proportion of adhesome proteins had high relative abundance in the nuclear fraction (displayed as peaks of high scaled intensities in the nuclear fraction silhouettes in **Reviewer Fig. 4a**). Indeed, plotting only high-stringency nuclear meta-adhesome proteins reveals that the majority of these have high relative abundance in the nucleus, with a much lower proportion detected at relatively low levels in the nucleus (**Reviewer Fig. 4b**). These data indicate that the subset of adhesome proteins detected at relatively high levels in the nucleus is represented predominantly by the high-stringency nuclear meta-adhesome proteins defined in our analyses. Moreover, separate plotting of biological replicate experiments shows that this distribution of adhesion protein abundance, featuring a population of high-relative-abundance adhesome proteins in the nucleus, was

consistently observed across all five independent biological replicate experiments (**Reviewer Fig. 4a,b**). To highlight this observation, a condensed representation of this analysis (omitting the breakdown of individual replicate density distributions shown in Reviewer Fig. 4) has been added as a new **Supplementary Fig. 3a** and described in the text (**Results, p. 8**).

Reviewer Figure 4 Relative quantification of meta-adhesome proteins across subcellular fractions. **(a)** Distribution of quantification of all detected meta-adhesome proteins. **(b)** Distribution of quantification of high-stringency nuclear meta-adhesome proteins. Data from all independent biological replicates (rep.) are shown separately. Black bars, 95% confidence interval; silhouettes, probability density.

The data presented in Supplementary Data 3 are useful as they are linked to adhesome proteins in the clustered heatmap in Fig. 2f, which is based on scaled intensities. We used scaled intensities to enable an intuitive relative comparison of protein abundance in different subcellular locations. We apologise for the lack of detail about the scaling approach; details of how the scaling was applied have now been added to the text (**Methods, p. 26**) and to the figure legend for Fig. 2. Details of protein LFQ intensities, from which the relative scaled values in Supplementary Data 3 were derived, are provided in Supplementary Data 2.

To convince the reader further the authors go on to perform microscopy, but only on a few proteins, where are these proteins on figure 2g?

Response and action taken

We chose to verify and quantify nuclear localisation of several adhesome proteins key to our findings in this manuscript using confocal microscopy. As mentioned earlier, FAK visibly and strongly stains focal adhesions because in that locale it exists in large protein clusters (see our report in DOI: 10.15252/emj.2020104743); this is not the case in the nucleus, where it is more challenging to visualise FAK by immunofluorescence using standard confocal microscopy. We have, however, been able to detect nuclear FAK in nuclear puncta (of unknown substructure and content) using super-resolution microscopy (see **Reviewer**

Fig. 5 below). We observed Myc-tagged FAK within the nucleus and, strikingly, at the nuclear periphery within putative nuclear lamina fenestrations, likely a snapshot of the dynamic nucleocytoplasmic transport of FAK across the nuclear envelope. This provides further strong support of the nuclear localisation of one of a subset of adhesion proteins. Because we do not understand the nature of the FAK-containing puncta and it is not possible to quantify by super-resolution imaging, we have not included these images at the present time. Screening all adhesome proteins for nuclear localisation would require that we evaluate/optimize high-quality antibodies that work in SIM microscopy or tag the proteins individually, and such a task is well beyond the scope of the current manuscript.

Reviewer Figure 5 Structured illumination microscopy (SIM) super-resolution imaging of an SCC cell. Myc-tagged FAK was expressed in SCC FAK^{-/-} cells and immunostained using anti-Myc tag antibody (which does not recognise endogenous c-Myc) (green); lamin A/C (magenta) underlies the inner nuclear membrane. Images were acquired on an N-SIM super-resolution microscope using a 100× 1.49NA lens and refractive index-matched immersion oil. For each focal plane, 15 images (5 phases, 3 angles) were captured. White arrowheads indicate examples of FAK localisation at the nuclear lamina in putative fenestrations devoid of lamin A/C staining; white arrows indicate examples of FAK localisation within the nucleus. Scale bars (top images), 1 μm; orthogonal section scale bar, 2 μm.

In summary, I still think the authors have a little work to do on the way in which their data are portrayed to convince the reader that the nuclear meta-adhesome proteins are nuclear and not present in their nuclear enriched fractions by virtue of contamination from other subcellular compartments.

We hope our extensive additional analyses and changes to the manuscript have addressed the remaining reviewers' comments.

Minor corrections:

1. The figure legend 2g points to another t-SNE plot as the supplementary fig. 2e. This should read 2d

Action taken

This callout has been removed from the figure legend.

2. It is not clear what the units are in supplementary table 2

Response and action taken

Label-free quantification (LFQ) is a measure of relative protein abundance that is reported as LFQ intensities (arbitrary units). This has been clarified in the headers in Supplementary Data 2 and 5 (formerly Supplementary Tables 2 and 5).

Additional notes

We have replaced the immunoblot images shown in Fig. 2b with longer exposures of the same membranes from the same experiment.

We have replaced the immunoblots shown in Fig. 4a,b with corresponding immunoblots from another independent biological replicate of the experiment for which the chromatin fraction loading control (histone H3) was probed on the same membrane as FAK. The results are the same as for the previous version.

We have replaced the immunoblots for the nuclear fraction loading controls (lamin-A/C and HP1 α/β) shown in Supplementary Fig. 4b with the correct loading control immunoblots from the same gel and the same experiment as the FAK immunoblot.

REVIEWERS' COMMENTS

Reviewer #1 (Remarks to the Author):

The authors have done an excellent job in responding to my concerns/suggestions.

My only remaining comment is that they should place reviewer figure 2 in the supplementary data to demonstrate the performance of their SVM Optimisation

REVIEWERS' COMMENTS

Reviewer #1 (Remarks to the Author):

The authors have done an excellent job in responding to my concerns/suggestions. My only remaining comment is that they should place reviewer figure 2 in the supplementary data to demonstrate the performance of their SVM Optimisation

Response and action taken

We are very pleased that the reviewer is satisfied with our revisions to the manuscript. We have included plots summarising SVM performance optimisation as a new **Supplementary Fig. 2e**, as requested by the reviewer.

Additional notes

We have corrected the positioning of the asterisks indicating statistical significance in Fig. 4h. We have included the exact P values corresponding to the asterisks in Fig. 4h in a new **Supplementary Data 9**.